# A gustatory receptor tuned to the steroid plant hormone brassinolide in *Plutella xylostella* (Lepidoptera: Plutellidae)

**Ke Yang**[1,2], **Xin-Lin Gong**[1,2], **Guo-Cheng Li**[1,2], **Ling-Qiao Huang**[1], **Chao Ning**[1,2], **Chen-Zhu Wang**[1,2]*

[1]State Key Laboratory of Integrated Management of Pest Insects and Rodents, Institute of Zoology, Chinese Academy of Sciences, Beijing, China; [2]CAS Center for Excellence in Biotic Interactions, University of Chinese Academy of Sciences, Beijing, China

**Abstract** Feeding and oviposition deterrents help phytophagous insects to identify host plants. The taste organs of phytophagous insects contain bitter gustatory receptors (GRs). To explore their function, the GRs in *Plutella xylostella* were analyzed. Through RNA sequencing and qPCR, we detected abundant *PxylGr34* transcripts in the larval head and adult antennae. Functional analyses using the *Xenopus* oocyte expression system and 24 diverse phytochemicals showed that PxylGr34 is tuned to the canonical plant hormones brassinolide (BL) and 24-epibrassinolide (EBL). Electrophysiological analyses revealed that the medial sensilla styloconica of 4[th] instar larvae are responsive to BL and EBL. Dual-choice bioassays demonstrated that BL inhibits larval feeding and female oviposition. Knock-down of PxylGr34 by RNAi attenuates the taste responses to BL, and abolishes BL-induced feeding inhibition. These results increase our understanding of how herbivorous insects detect compounds that deter feeding and oviposition, and may be useful for designing plant hormone-based pest management strategies.

**\*For correspondence:**
czwang@ioz.ac.cn

**Competing interests:** The authors declare that no competing interests exist.

## Introduction

Many phytophagous insects have evolved to select a limited range of host plants. Understanding the ultimate and proximate mechanisms underlying this selection strategy is a crucial issue in the field of insect–plant interactions. Insects' decisions to feed and oviposit are mainly based on information carried via chemosensory systems (*Schoonhoven et al., 2005*). Insects discriminate among potential hosts after perceiving a combination of lineage-specific and more ubiquitous chemicals synthesized by plants. Some nutrients are ubiquitous across plant taxa. For example, various sugars and amino acids are feeding stimulants for the majority of herbivorous insects. Secondary plant metabolites occur in certain plant taxa at much higher concentrations than in others, and therefore are of greater significance in host-plant selection by insects (*Jermy, 1966*). They are usually deterrent or 'bitter' compounds for many phytophagous insects, except for some specialized species that use them as token stimuli (*Schoonhoven et al., 2005*).

Phytophagous insects have sophisticated taste systems to recognize deterrent or stimulant compounds, which direct their feeding and oviposition behavior (*Yarmolinsky et al., 2009*). The taste sensilla of insects are mainly situated on the mouthparts, tarsi, ovipositor, and antennae (*Bernays and Chapman, 1994*). These sensilla take the form of hairs or cones with a terminal pore in the cuticular structure, and often contain the dendrites of three or four gustatory sensory neurons (GSNs). The axons of GSNs synapse directly onto the central nervous system. Analyses using the tip-recording technique for taste sensilla led to the discovery of a 'deterrent neuron' in the larvae of *Bombyx mori* (*Ishikawa, 1966*) and *Pieris brassicae* (*Ma, 1969*). Since then, GSNs coding for

**eLife digest** Plant-eating insects use their sense of taste to decide where to feed and where to lay their eggs. They do this using taste sensors called gustatory receptors which reside in the antennae and legs of adults, and in the mouthparts of larvae. Some of these sensors detect sugars which signal to the insect that the plant is a nutritious source of food. While others detect bitter compounds, such as poisons released by plants in self-defense.

One of the most widespread plant-eating insects is the diamondback moth, which feeds and lays its eggs on cruciferous vegetable crops, like cabbage, oilseed rape and broccoli. Before laying its eggs, female diamondback moths pat the vegetable's leaves with their antennae, tasting for the presence of chemicals. But little was known about the identity of these chemicals.

Cabbages produce large amounts of a hormone called brassinolide, which is known to play a role in plant growth. To find out whether diamondback moths can taste this hormone, Yang et al. examined all their known gustatory receptors. This revealed that the adult antennae and larval mouthparts of these moths make high levels of a receptor called PxylGr34.

To investigate the role of PxylGr34, Yang et al. genetically modified frog eggs to produce this receptor. Various tests on these receptors, as well as receptors in the mouthparts of diamondback larvae, showed that PxylGr34 is able to sense the hormone brassinolide. To find out how this affects the behavior of the moths, Yang et al. investigated how adults and larvae responded to different levels of the hormone. This revealed that the presence of brassinolide significantly decreased both larval feeding and the amount of eggs laid by adult moths.

Farmers already use brassinolide to enhance plant growth and protect crops from stress. These results suggest that the hormone might also help to shield plants from insect damage. However, more research is needed to understand how this hormone acts as a deterrent. Further studies could improve understanding of insect behavior and potentially identify more chemicals that can be used for pest control.

secondary plant metabolites have been identified in maxillary sensilla in larvae and tarsal sensilla of adults of many Lepidopteran species (*Glendinning et al., 2002*; *Zhou et al., 2009*). However, the molecular basis of these deterrent neurons remains unclear.

Gustatory receptors (GRs) expressed in the dendrites of GSNs determine the selectivity of the response of GSNs (*Thorne et al., 2004*; *Wang et al., 2004*). Since the first insect GRs were identified in the model organism *Drosophila melanogaster* (*Clyne et al., 2000*), the function of some of its bitter GRs have been revealed (*Dweck and Carlson, 2020*; *Freeman and Dahanukar, 2015*). Five *D. melanogaster* GRs (Gr47a, Gr32a, Gr33a, Gr66a, and Gr22e) are involved in sensing strychnine (*Lee et al., 2010*; *Lee et al., 2015*; *Moon et al., 2009*; *Poudel et al., 2017*). Nicotine-induced action potentials are dependent on Gr10a, Gr32a, and Gr33a (*Rimal and Lee, 2019*). Gr8a, Gr66a, and Gr98b function together in the detection of L-canavanine (*Shim et al., 2015*). Gr33a, Gr66a, and Gr93a participate in the responses to caffeine and umbelliferone (*Lee et al., 2009*; *Moon et al., 2006*; *Poudel et al., 2015*). Gr28b is necessary for avoiding saponin (*Sang et al., 2019*). However, only a few studies have focused on the function of bitter GRs in phytophagous insects. PxutGr1, a bitter GR in *Papilio xuthus*, was found to respond specifically to the oviposition stimulant, synephrine (*Ozaki et al., 2011*). In recent studies, the insect bitter GRs BmorGr16, BmorGr18, and BmorGr53 showed response to coumarin and caffeine in vitro, and coumarin was found to have a feeding deterrent effect on *B. mori* larvae (*Kasubuchi et al., 2018*); another bitter GR in *B. mori*, Gr66, was reported to be responsible for the mulberry-specific feeding preference of silkworms (*Zhang et al., 2019*). Although some progress has been made in determining the functions of such receptors in *Drosophila* and a few herbivorous insects, we still lack basic mechanistic information about the functions of bitter GRs from most herbivorous insect lineages.

*Plutella xylostella* (L.) is the most widespread Lepidopteran pest species, causing losses of US$ 4–5 billion per year (*You et al., 2020*; *Zalucki et al., 2012*). It has developed resistance to the usual insecticides because of its short life cycle (14 days) (*Furlong et al., 2013*). *P. xylostella* mainly selects Brassica species as its host plants, and its females pat the leaf surfaces with their antennae before egg laying (*Qiu et al., 1998*). This behavior may be related to certain chemical components in leaves

of *Brassica* species, including sugars, sugar alcohols, amino acids, amines, glucosinolates, and plant hormones. Among these compounds, sinigrin and brassinolide (BL) have relatively higher concentrations in Brassica than in many other plant species (*Fahey et al., 2001*; *Lv et al., 2014*). Sinigrin is known to be a feeding/oviposition stimulant for *P. xylostella* (*Gupta and Thorsteinson, 1960*). The medial sensilla styloconica in the maxillary galea of *P. xylostella* larvae contain a GSN sensitive to sinigrin and other glucosinolates (*van Loon et al., 2002*). BL is a ubiquitous plant hormone that has been widely studied in relation to its role in plant growth and development (*Clouse and Sasse, 1998*), but little is known about its effects on the behavior of phytophagous insects.

To uncover the molecular basis of the perception of feeding/oviposition stimulants and deterrents by *P. xylostella*, we re-examined all the GRs reported in previous studies on this insect. Through transcriptome analysis and qPCR, we identified one bitter GR (PxylGr34) highly expressed in the larval head and the adult antennae. We functionally analyzed this GR with the *Xenopus* oocyte expression system and RNAi, and found that PxylGr34 is tuned to BL as a feeding and oviposition deterrent in *P. xylostella*.

## Results

### Identification of *PxylGr34*, a highly expressed GR gene in *P. xylostella*

To search for candidate GRs that may be involved in host selection by *P. xylostella*, we searched for candidates among those that had been annotated in the *P. xylostella* genome (*Engsontia et al., 2014*; *You et al., 2013*), the transcriptome (*Yang et al., 2017*), and the *P. xylostella* GRs deposited in GenBank (*Supplementary file 1*). We obtained 79 annotated GRs (69 GRs from the genome, 7 GRs from the transcriptome, and 3 GRs from GenBank). After removing repetitive sequences and deleting paired sequences with amino acid identity greater than 99%, we validated 67 of the GR sequences (61 GRs from the genome, 3 GRs from the transcriptome, and 3 GRs from GenBank) (*Figure 1* and *Supplementary file 1*). On this basis, we first analyzed the transcriptomes of the antennae, forelegs (only tibia and tarsi), and head (without antennae) of adults, and the mouthparts of 4th instar larvae although the genomic data and antennae transcriptome of *P. xylostella* had been reported previously (*Yang et al., 2017*; *You et al., 2013*). Next, we looked specifically for candidates that were highly expressed in these chemosensory organs by calculating the transcripts per million (TPM) values of these 67 GRs (*Figure 2A*). Those GRs with high expression levels in both adult and larval taste organs were considered to be candidates for those with key roles in the host-plant selection of this species. Intriguingly, the TPM value of *PxylGr34*, which clustered with bitter GRs, was much higher than those of other GR genes in the antennae, head and forelegs of adults, as well as the mouthparts of larvae (*Figure 2A*). *PxylGr34* was originally annotated from genomic data (*Engsontia et al., 2014*), and then detected in the antennal transcriptomic data of *P. xylostella* (named as '*PxylGr2*' in *Yang et al., 2017*). However, both studies provided only its partial coding sequences (*Figure 1—figure supplement 1*). Based on our transcriptomic data, we obtained the full-length coding sequence of *PxylGr34* through gene cloning and Sanger sequencing (*Figure 1—figure supplement 1*). The protein encoded by *PxylGr34* is typical of most GRs with seven transmembrane domains, and a full open reading frame (ORF) of 418 amino acids (*Figure 1—figure supplement 2*).

### High PxylGr34 transcript levels in the larval head and adult antennae

To further confirm the expression patterns of *PxylGr34* in the larvae and adults, we detected its relative transcript levels in different tissues of adults and the 4th instar larvae of *P. xylostella* using quantitative real-time PCR (qPCR). The larvae eat the most in the 4th instar and can forage around the plant more easily (*Harcourt, 1957*). The high levels of *PxylGr34* transcripts were detected in the larval head. They were also detected in the larval thoracic legs and gut (*Figure 2B*). In the adults, *PxylGr34* transcripts were restricted to the antennae (*Figure 2C*).

### BL and EBL induced a strong response in the oocytes expressing PxylGr34

We used the *Xenopus laevis* oocyte expression system and two-electrode voltage-clamp recording to study the function of PxylGr34. Among 24 tested phytochemicals belonging to sugars, sugar

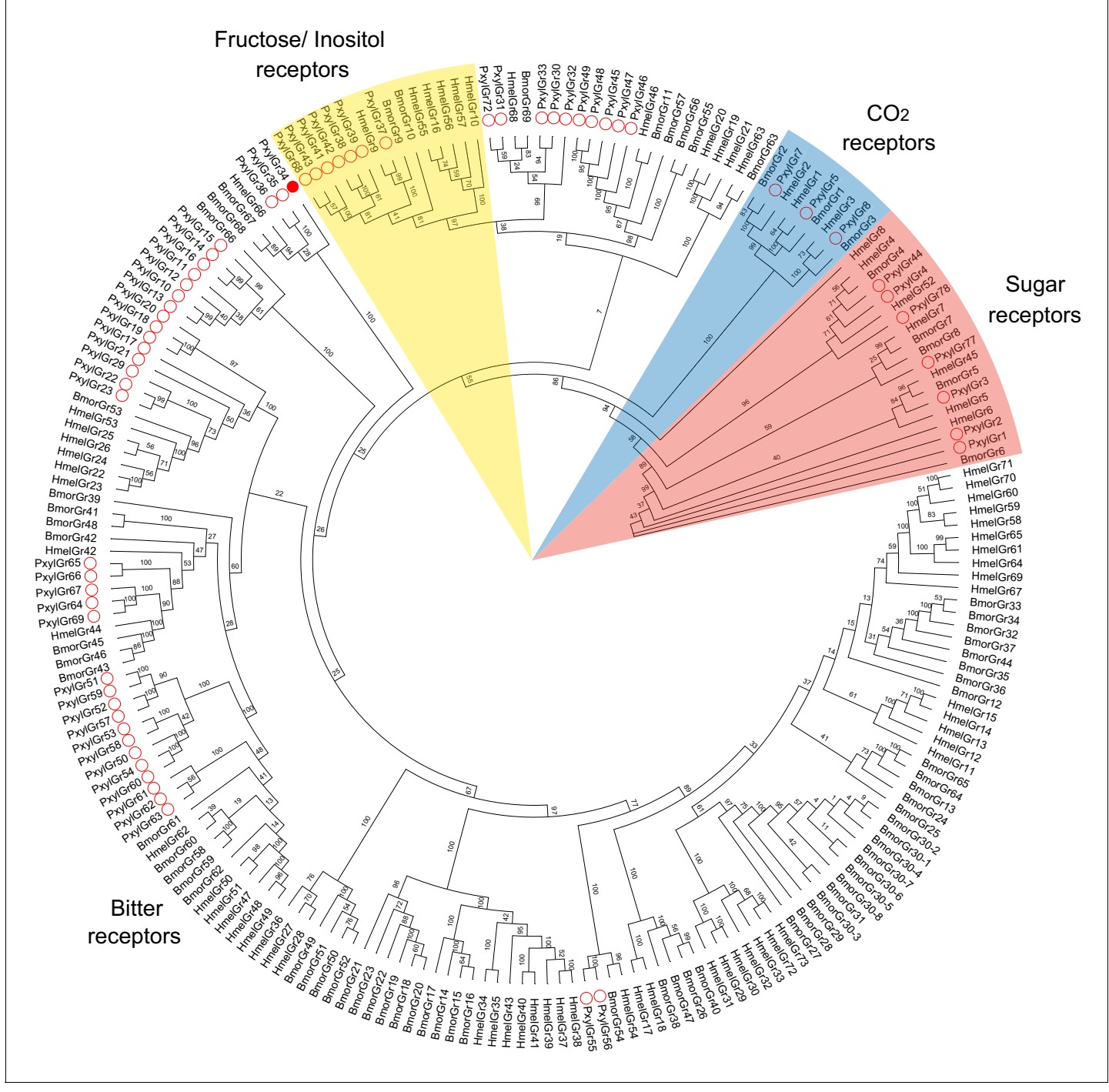

**Figure 1.** Phylogenetic tree of gustatory receptors (GRs). Amino acid sequences are based on previously reported GRs. Bootstrap values are based on 1000 replicates. Abbreviations: Hmel, *Heliconius melpomene*; Bmor, *Bombyx mori*; Pxyl, *Plutella xylostella*. ○, GRs of *P. xylostella*; •, PxylGR34. The online version of this article includes the following source data and figure supplement(s) for figure 1:

**Source data 1.** Amino acid sequences of GRs in *Plutella xylostella*.

**Figure supplement 1.** Alignment of amino acid sequences of PxylGr34 from genomic data (*Engsontia et al., 2014*), transcriptomic data (*Yang et al., 2017*), and this study.

**Figure supplement 2.** Secondary structure prediction of PxylGr34.

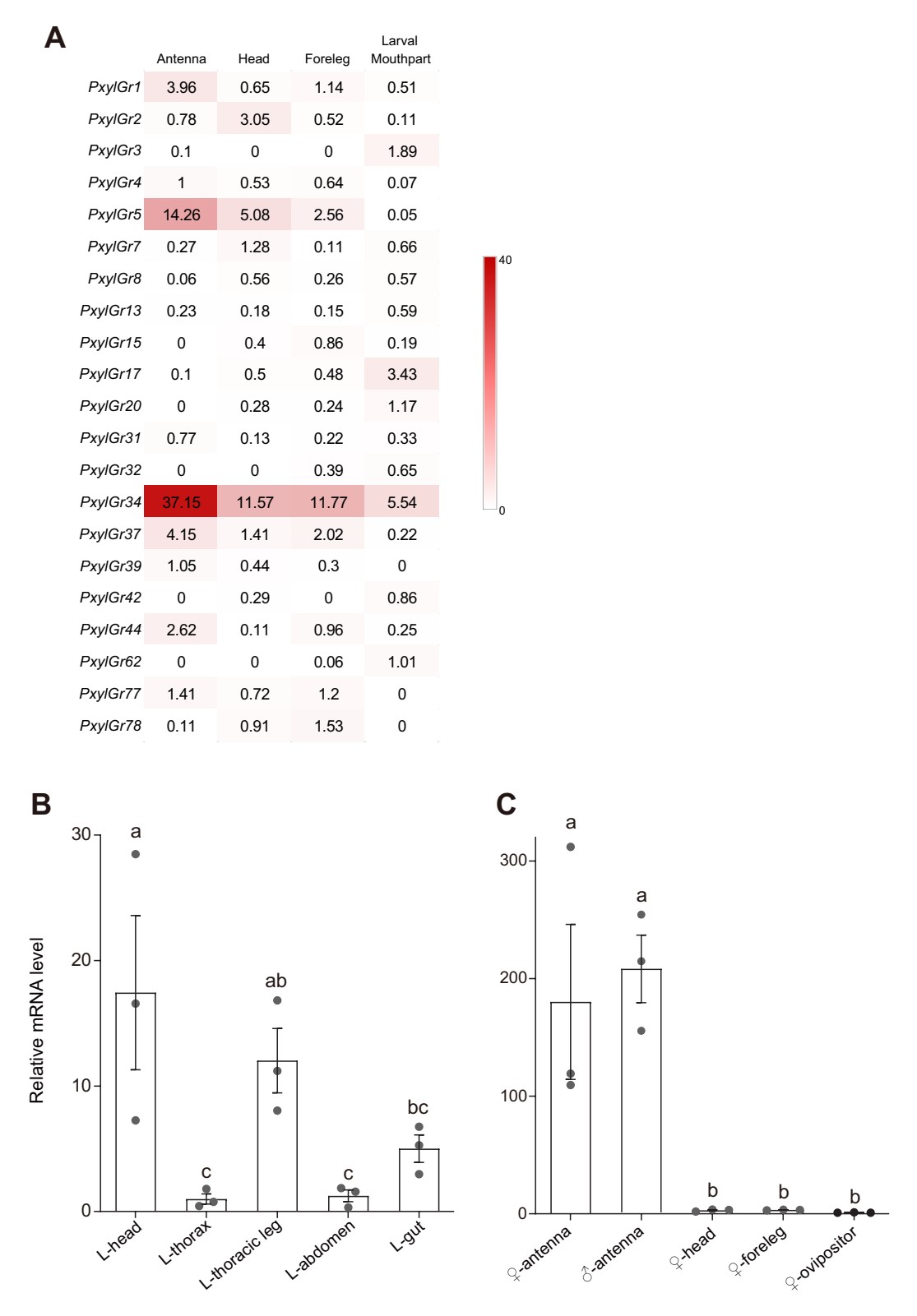

**Figure 2.** Tissue expression pattern of gustatory receptors (GRs) in *Plutella xylostella* as determined by Illumina read-mapping and qPCR analysis. (**A**) Transcripts per million (TPM) value of each GR is indicated in box. Color scales were generated using Microsoft Excel. Antenna, head, and foreleg were from the moth; larval mouthpart was from 4th instar larvae. The GRs that undetectable in the TPM analysis were not listed. Relative *PxylGr34* transcript

*Figure 2 continued on next page*

*Figure 2 continued*

levels in (B) 4th instar larval tissues and (C) mated moth tissues of *Plutella xylostella* as determined by qPCR. Data are mean ± SEM. *n* = 3 replicates of 40–200 tissues each. For 4th instar larvae, p=0.0004; for the moth, p<0.0001 (one-way ANOVA, Tukey's HSD test).

The online version of this article includes the following source data for figure 2:

**Source data 1.** Source data for *Figure 2B and C*.

alcohols, amino acids, amines, glucosinolates, and plant and insect hormones (Key Resources Table), BL induced a strong response in the oocytes expressing PxylGr34, as did its racemate EBL at a concentration of $10^{-4}$ M (*Figure 3A* and *Figure 3B*). The currents induced by BL increased from the lowest threshold concentration of $10^{-4}$ M in a dose-dependent manner (*Figure 3C* and *Figure 3D*). Oocytes expressing PxylGr34 showed weak responses to methyl jasmonate and allyl isothiocyanate, but no response to 20-hydroxyecdysone and other tested compounds (*Figure 3A* and *Figure 3B*). As negative controls, the water-injected oocytes failed to respond to any of the tested chemical stimuli (*Figure 3—figure supplement 1*).

### The larval medial sensilla styloconica exhibited vigorous responses in *P. xylostella* to BL and EBL

Next, using the tip-recording technique, we examined whether any gustatory sensilla in the mouthparts of larvae of *P. xylostella* could respond to BL and EBL. Of the two pairs of sensilla styloconica in the maxillary galea of 4th instar larvae, the lateral sensilla styloconica had no response to BL and EBL (*Figure 4A,B,C and D*); the medial sensilla styloconica exhibited vigorous responses to BL and EBL at $3.3 \times 10^{-4}$ M, and the spike amplitudes induced by BL and EBL were about the same (*Figure 4A,B,C and D*). As previously reported, the medial sensilla styloconica also exhibited vigorous responses to sinigrin. However, the spike amplitudes induced by sinigrin were larger than those induced by BL and EBL (*Figure 4E and F*). These results suggest that BL and EBL activate the same neuron, while sinigrin activates a different neuron in the sensillum. The medial sensilla styloconica showed a dose-dependent response to BL, although the testing concentrations were limited because of the low solubility of BL in water (highest concentration approximately $3.3 \times 10^{-4}$ M) (*Figure 5*).

### BL and EBL-induced feeding deterrence effect on *P. xylostella* larvae

We further tested the effects of BL and EBL on the larval feeding behavior of *P. xylostella* on pea leaves in the dual-choice leaf disc assay. In a dual-choice feeding test with 4th instar larvae, the feeding areas of larvae were significantly smaller on the leaf discs treated with BL and EBL at concentrations of $10^{-4}$ M and above than on the control leaf discs. In addition, the feeding preference of larvae to BL and EBL tended to decrease with increasing BL and EBL concentrations (*Figure 6* and *Figure 6—figure supplement 1*), indicating that both BL and EBL function as feeding deterrents to *P. xylostella* larvae.

### BL-induced oviposition deterrence to *P. xylostella* females

We also tested the effects of BL on the female ovipositing behavior of *P. xylostella*. In a dual-choice oviposition test with mated females, significantly fewer eggs were laid on the sites treated with BL at $10^{-4}$ M, $10^{-3}$ M, and $10^{-2}$ M than on the control sites. In addition, the oviposition preference tended to decrease as the BL concentrations increased (*Figure 7*).

### PxylGr34 siRNA-treated *P. xylostella* larvae show attenuated taste responses to BL and alleviated feeding deterrent effect of BL

To clarify whether PxylGr34 mediates the electrophysiological and behavioral responses of *P. xylostella* larvae to BL *in vivo*, we tested the effect of siRNA targeting *PxylGr34* on the responses of medial sensilla styloconica and the feeding behavior of the 4th instar larvae. The relative transcript level of *PxylGr34* in the head of larvae treated with PxylGr34 siRNA was half that in the head of larvae treated with green fluorescent protein (GFP) siRNA or ddH$_2$O (*Figure 8A*). This confirmed that feeding with siRNA is an effective method for RNAi-inhibition of *PxylGr34* in the larval head.

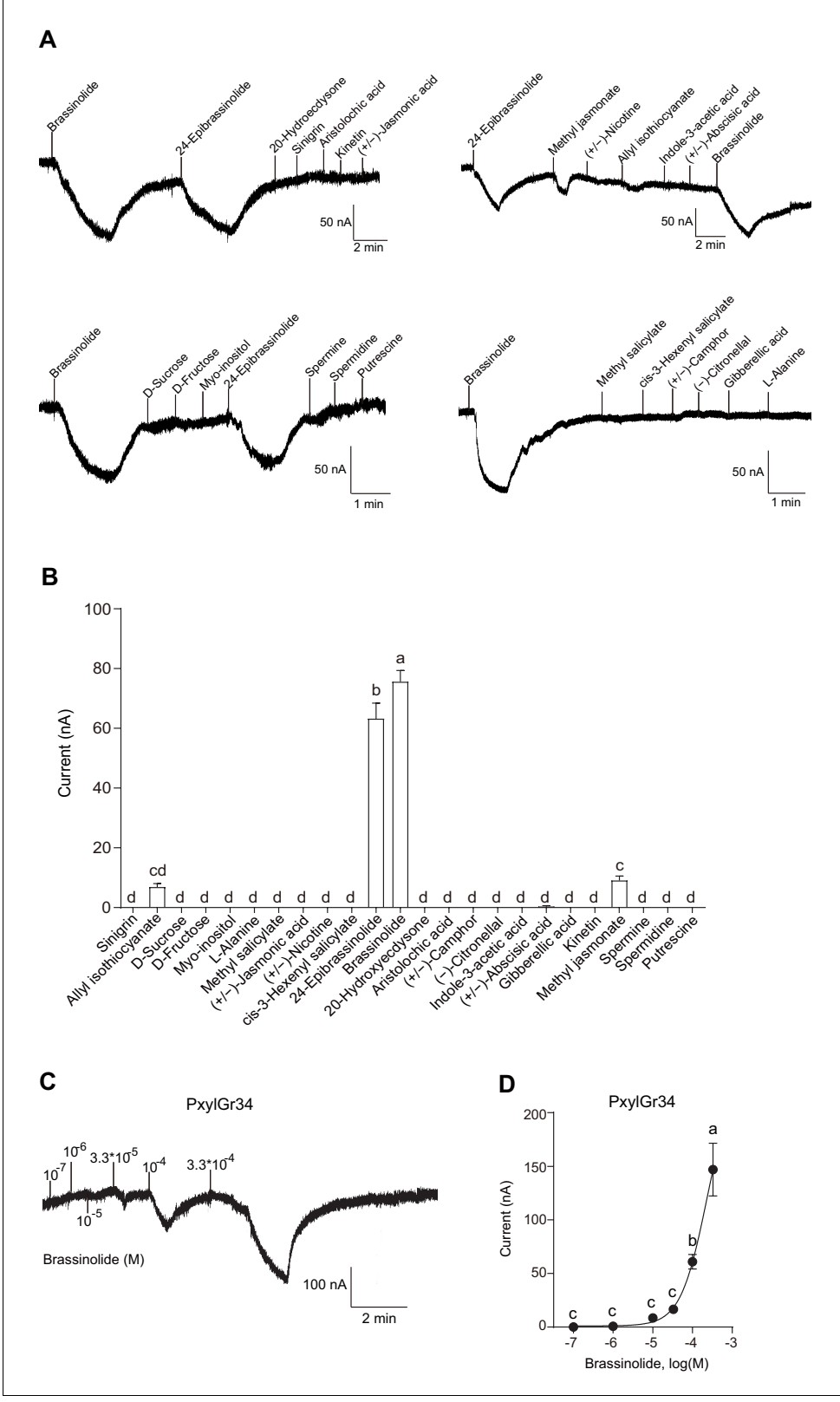

**Figure 3.** Brassinolide and 24-epibrassinolide induced a strong response in the oocytes expressing PxylGr34. (**A**) Representative inward current responses of *Xenopus* oocytes expressing PxylGr34 in response to ligands at $10^{-4}$ M. (**B**) Response profiles of *Xenopus* oocytes expressing PxylGr34 in response to ligands at $10^{-4}$ M. Data are mean ± SEM. *n* = 7 replicates of cells. p<0.0001 (one-way ANOVA, Tukey's HSD test). (**C**) Representative inward
*Figure 3 continued on next page*

*Figure 3 continued*

current responses of *Xenopus* oocytes expressing PxylGr34 in response to BL at a range of concentrations. (**D**) Response profiles of *Xenopus* oocytes expressing PxylGr34 in response to BL at a range of concentrations. Data are mean ± SEM; *n* = 6–8 replicates of cells. p<0.0001 (one-way ANOVA, Tukey's HSD test).

The online version of this article includes the following source data and figure supplement(s) for figure 3:

**Source data 1.** Source data for *Figure 3B and D*.

**Figure supplement 1.** Two-electrode voltage-clamp recordings of *Xenopus* oocytes injected with distilled water and stimulated with test compounds.

To test the effects of RNAi-inhibition of *PxylGr34* on the taste responses, the medial sensilla styloconica from siRNA-treated larvae were subjected to a tip-recording analysis as described above, with $3.3 \times 10^{-4}$ M BL or water. As shown in *Figure 8*, although the medial sensilla styloconica of PxylGr34 siRNA-treated larvae still showed some response to BL, the frequency of spikes to BL elicited in the medial sensilla styloconica of the PxylGr34 siRNA-treated larvae was decreased (*Figure 8B* and *Figure 8C*).

To test the effects of RNAi-knockdown of *PxylGr34* on the feeding behavior of $4^{th}$ instar larvae, the siRNA-treated larvae were subjected to a dual-choice leaf disc feeding assay as described above, with leaf discs of pea treated with $10^{-4}$ M BL or untreated (control). As shown in *Figure 9*, both the water-treated larvae and the GFP siRNA-treated larvae preferred control leaf discs over those treated with BL, whereas the PxylGr34 siRNA-treated larvae showed no significant preference (*Figure 9*). Thus, the knock-down of PxylGr34 by RNAi attenuated the electrophysiological responses of the medial sensilla styloconica to BL, and alleviated the deterrent effect of BL on the feeding of *P. xylostella* larvae.

## Discussion

In this study, we identified the full-length coding sequence of *PxylGr34* from our transcriptome data, and found that this gene is highly expressed in the head of the $4^{th}$ instar larvae and in the antennae of females. Our results show that PxylGr34 is specifically tuned to BL and its racemate EBL, and that the medial sensilla styloconica of $4^{th}$ instar larvae have electrophysiological responses to BL and EBL. Our results also show that BL inhibits larval feeding and female oviposition of *P. xylostella*, and that knock-down of PxylGr34 by RNAi can attenuate the responses of sensilla to BL, and abolish the feeding inhibition effect of BL. This is the first study to show that an insect can detect and react to this steroid plant hormone. The results of the systematic functional analyses of the GR, the electrophysiological responses of the sensilla, behavioral assays, and behavioral regulation *in vivo* show that PxylGr34 is a bitter GR specifically tuned to BL. This receptor mediates the deterrent effects of BL on feeding and ovipositing behaviors of *P. xylostella*.

The larvae of Lepidopteran species have two pairs of gustatory sensilla (medial and lateral sensilla styloconica) located in the maxillae galea; these sensilla play a decisive role in larval food selection (*Dethier, 1937*; *Schoonhoven and van Loon, 2002*). Each sensillum usually contains four GSNs, of which one is often responsive to deterrents. *Ishikawa, 1966* described a 'deterrent neuron' in the medial sensillum styloconicum of silkworm, *Bombyx mori*, and showed that it responds to several plant alkaloids and phenolics (*Ishikawa, 1966*). Similar neurons have been found in other Lepidopteran species, but their profiles vary. For example, the tobacco hornworm, *Manduca sexta*, has a deterrent neuron in the medial sensillum styloconicum that responds to aristolochic acid, and another deterrent neuron in the lateral sensillum styloconicum that responds to salicin, caffeine, and aristolochic acid (*Glendinning et al., 2002*). The diversity of GRs facilitates the detection of, and discrimination among, a wide range of diverse taste stimuli, implying that different sets of bitter GRs are expressed in these neurons. In this study, we proved that one GSN in the medial maxillary sensillum styloconicum of the $4^{th}$ instar larvae of the diamondback moth responds to sinigrin (*van Loon et al., 2002*), and we identified one neuron responding to BL and EBL in the same sensillum. For *P. xylostella* larvae, sinigrin is a feeding stimulant whereas BL and EBL are feeding deterrents, together with sinigrin and BL/EBL have different spike amplitudes, we speculate that the GRs tuned to sinigrin and BL/EBL are located in different GSNs in the medial sensilla styloconica.

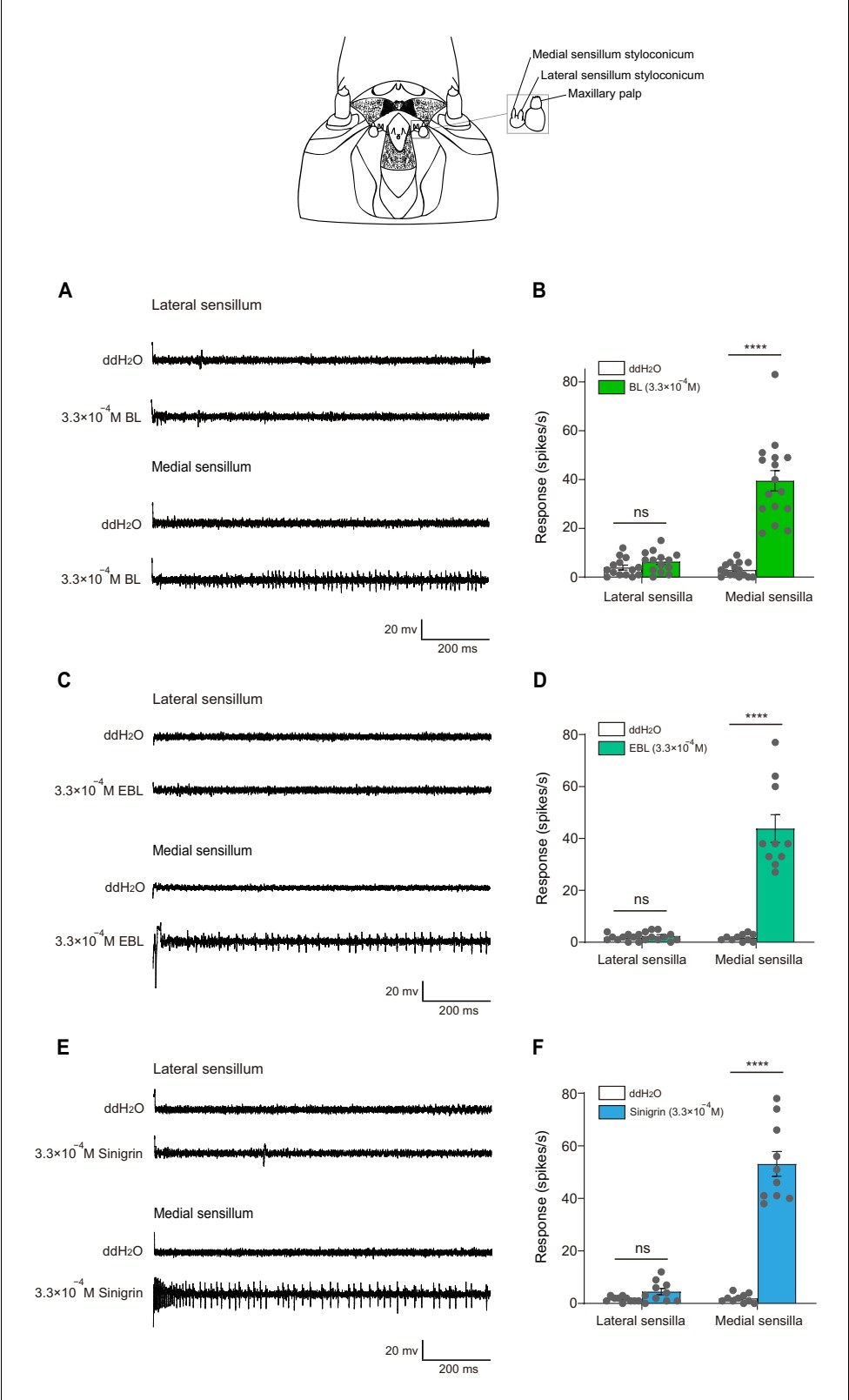

**Figure 4.** The larval medial sensilla styloconica exhibited vigorous responses in *P. xylostella* to brassinolide (BL) and 24-epibrassinolide (EBL). Typical electrophysiological recordings in response to water and BL (**A**), EBL (**C**), and sinigrin (**E**) at $3.3 \times 10^{-4}$ M for 1 s obtained by tip-recording from a neuron innervating the lateral and medial sensillum styloconica on maxillary galea of *P. xylostella* 4th instar larvae. Response profiles of the two lateral and medial sensilla styloconica on the maxilla of *P. xylostella* 4th instar larvae to water and BL (**B**), EBL (**D**), and sinigrin (**F**) at $3.3 \times 10^{-4}$ M. Data are

*Figure 4 continued on next page*

*Figure 4 continued*

mean ± SEM. For lateral sensilla styloconica, $n$ = 14, p=0.0905 to BL; $n$ = 10, p=0.3869 to EBL; $n$ = 10, p=0.0672 to sinigrin; for medial sensilla styloconica, $n$ = 16, p<0.0001 to BL; $n$ = 10, p<0.0001 to EBL; $n$ = 10, p<0.0001 to sinigrin. Data were analyzed by paired-samples $t$-test.

The online version of this article includes the following source data for figure 4:

**Source data 1.** Source data for *Figure 4B,D and F*.

The GR family is massively expanded in moth species, and most of the GRs are bitter GRs (*Cheng et al., 2017*). However, few studies have functionally characterized bitter GRs. In *D. melanogaster*, the loss of bitter GRs was found to eliminate repellent behavior in response to specific noxious compounds. For example, *Gr33a* mutant flies could not avoid non-volatile repellents like quinine and caffeine (*Moon et al., 2009*), and mutation of Gr98b impaired the detection of L-canavanine (*Shim et al., 2015*). When the bitter GR PxutGr1 of *P. xuthus* was knocked-down by RNAi, the oviposition behavior in response to synephrine was strongly reduced (*Ozaki et al., 2011*). Knock-out of the bitter GR BmorGr66 in *B. mori* larvae resulted in a loss of feeding specificity for mulberry (*Zhang et al., 2019*). In this study, BL and EBL induced a strong response in the oocytes expressing PxylGr34, knock-down of PxylGr34 in the larvae of *P. xylostella* eliminated the feeding deterrence of BL, indicating that this bitter GR is specifically tuned to BL and EBL, and mediates the aversive response of larvae to BL and related compounds. Although the 24 tested compounds represent a wide range of compound profiles, the ligands of PxylGr34 could be more than BL and EBL.

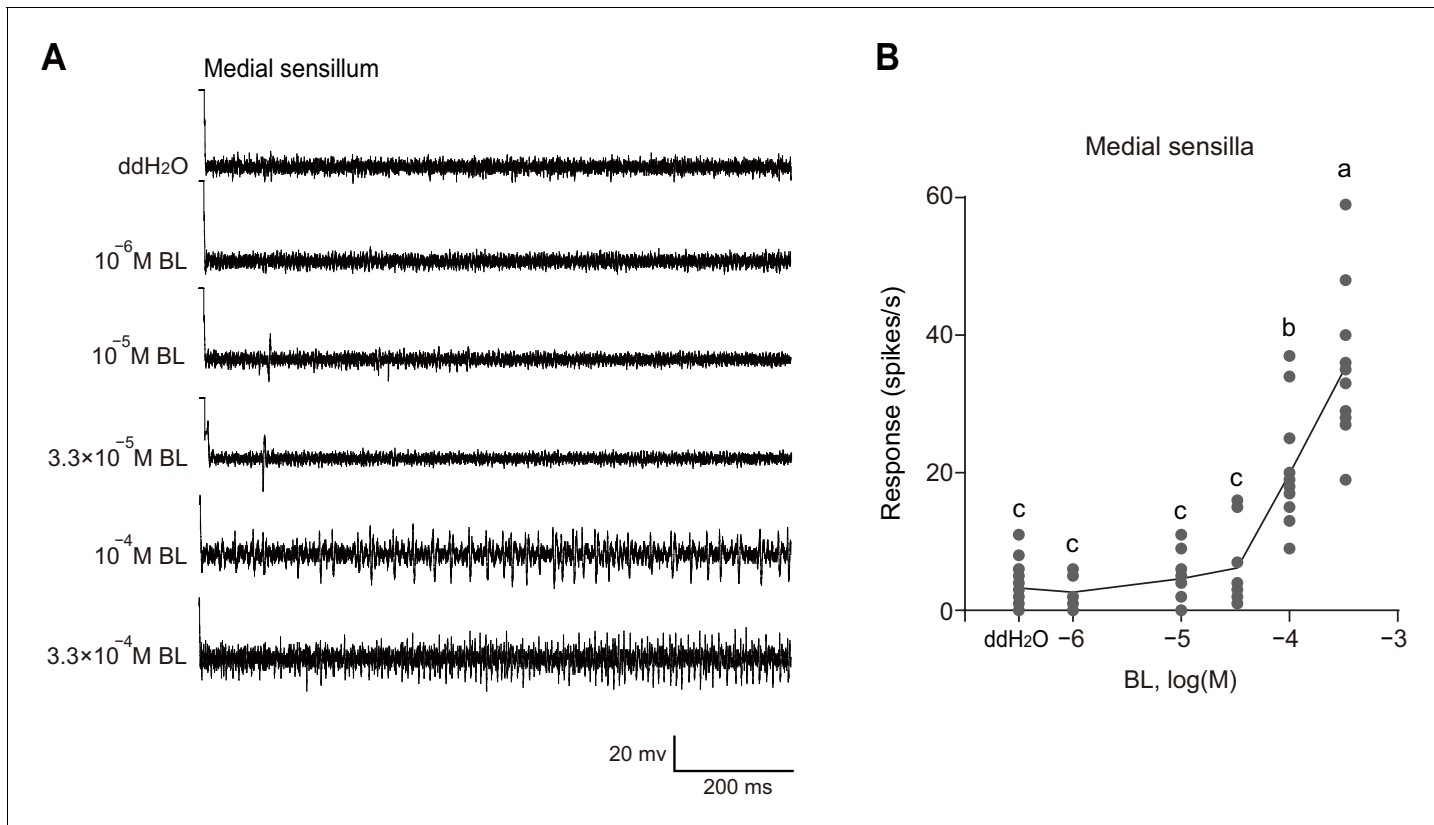

**Figure 5.** The medial sensilla styloconica showed a dose-dependent response to brassinolide (BL). (**A**) Typical electrophysiological recordings in response to water and BL at a series of concentrations for 1 s obtained by tip-recording from a neuron innervating the medial sensillum styloconicum on the maxillary galea of *P. xylostella* 4[th] instar larvae. (**B**) Response profiles of medial sensilla styloconica on maxilla of *P. xylostella* 4[th] instar larvae to water and BL at a series of concentrations. Data are mean ± SEM. $n$ = 8–14 replicates of larvae, p<0.0001 (one-way ANOVA, Tukey's HSD test).

The online version of this article includes the following source data for figure 5:

**Source data 1.** Source data for *Figure 5B*.

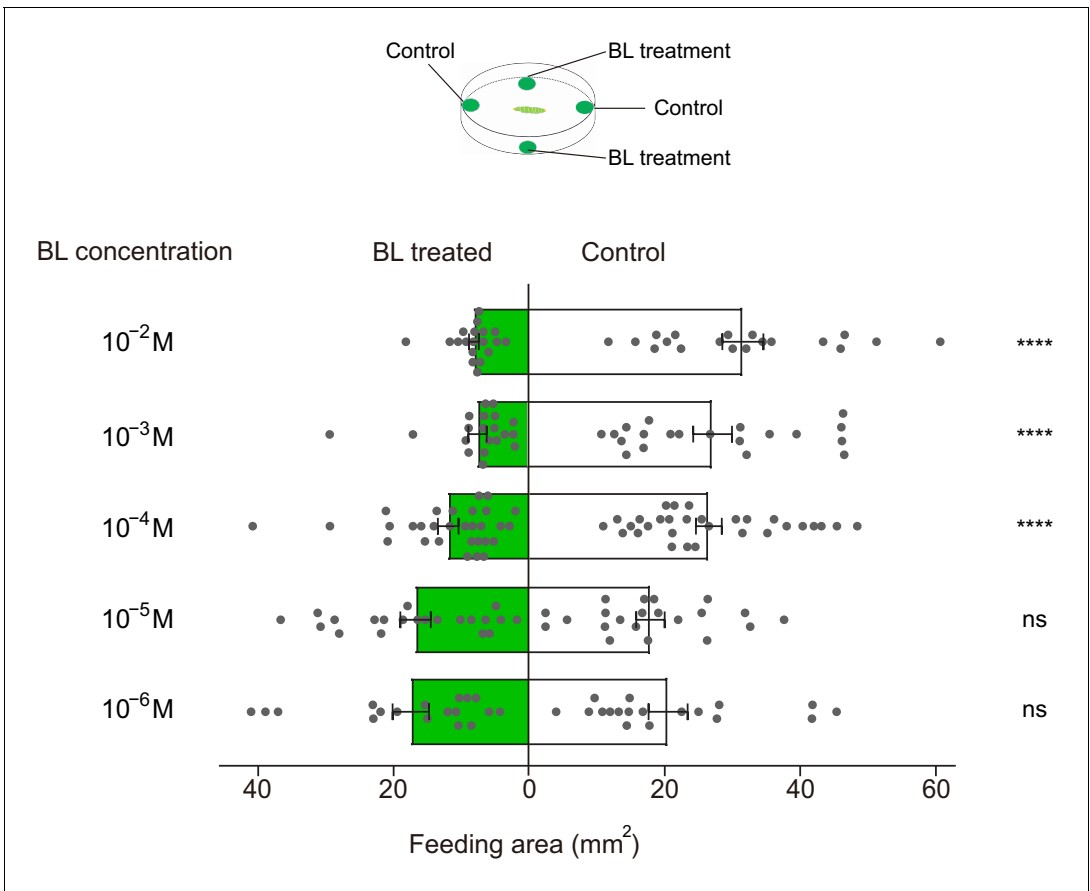

**Figure 6.** Brassinolide (BL)-induced feeding deterrence effect on *P.xylostella* larvae. Dual-choice feeding tests were conducted using pea leaf discs. Total feeding area of control (white bars) and BL-treated discs (green bars). BL was diluted in 50% ethanol to $10^{-6}$, $10^{-5}$, $10^{-4}$, $10^{-3}$, and $10^{-2}$ M. Data are mean ± SEM. For $10^{-6}$ M, $n = 18$, p=0.3947; for $10^{-5}$ M, $n = 21$, p=0.5967; for $10^{-4}$ M, $n = 30$, p<0.0001; for $10^{-3}$ M, $n = 20$, p<0.0001; for $10^{-2}$ M, $n = 19$, p<0.0001. Data were analyzed by paired-samples *t*-test.

The online version of this article includes the following source data and figure supplement(s) for figure 6:

**Source data 1.** Source data for *Figure 6*.

**Figure supplement 1.** 24-Epibrassinolide (EBL)-induced feeding deterrence effect on *P. xylostella* larvae.

**Figure supplement 1—source data 1.** Source data for *Figure 6—figure supplement 1*.

Given the high expression level of *PxylGr34* in the taste organs of *P. xylostella*, we could not rule out the possibility that this gene also functions together with other *GRs* to perceive other compounds.

BL was the first brassinosteroid (BR) hormone to be discovered in plants. It was first isolated and identified from a crude extract of pollen from oilseed rape (*Brassica napus* L.), and was found to induce rapid elongation of pinto bean *Phaseolus vulgaris* internodes distinct from gibberellin-mediated stem elongation (*Mitchell et al., 1970*; *Grove et al., 1979*). Almost all plant tissues contain BRs, and they function to promote elongation and stimulate cell division, participate in vascular differentiation and fertilization, and affect senescence (*Clouse and Sasse, 1998*). As a $C_{28}$ BR, BL exhibits the highest activity among all BRs and is distributed widely in the plant kingdom, along with other biosynthetically related compounds (*Clouse and Sasse, 1998*). Exogenous application of BL and its analog 24-epibrassinolide (EBL) to plants has been shown to increase their stress resistance (*Clouse and Sasse, 1998*).

Plant hormones, although generally found in small amounts and rarely toxic, play a key role in regulating plant growth, development, and resistance to biotic and abiotic stresses (*Bari and Jones, 2009*; *Krouk et al., 2011*; *Wu and Baldwin, 2010*). Jasmonic acid, salicylic acid, ethylene, and abscisic acid have been shown to be involved in priming plant defense responses against herbivorous insects and plant pathogens by activating related signal transduction pathways and changing the

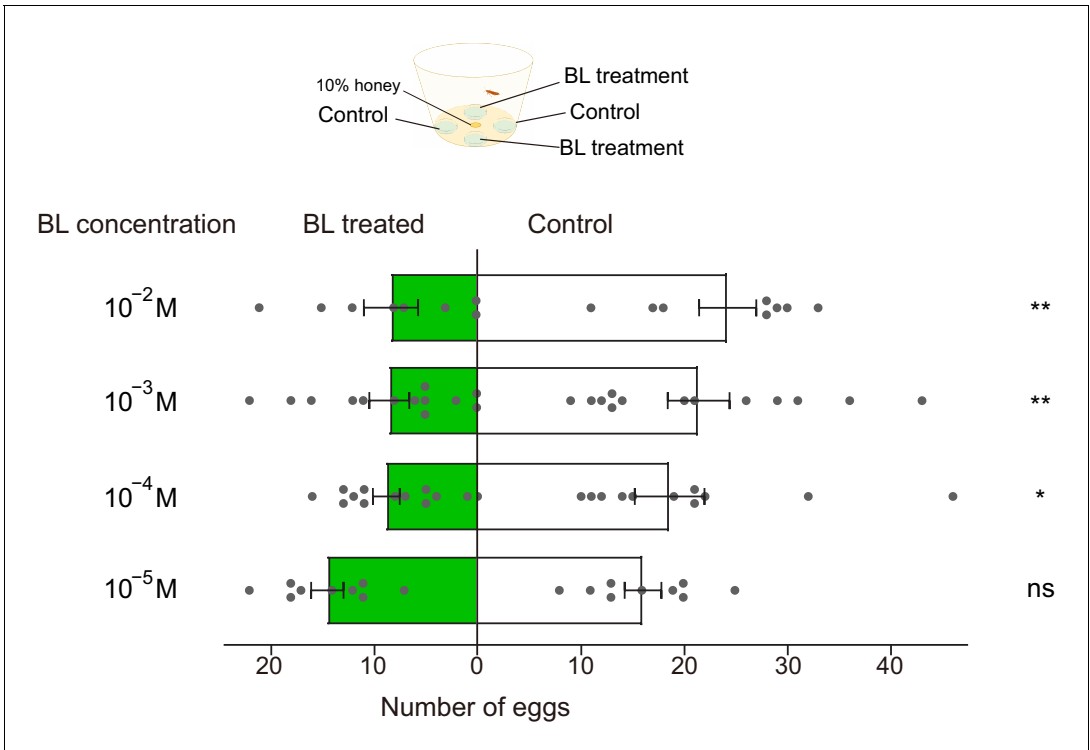

**Figure 7.** Brassinolide (BL)-induced oviposition deterrence to *P. xylostella* females. Dual-choice oviposition tests were conducted using plastic film coated with cabbage leaf juice. Both control (50% ethanol) and BL (diluted in 50% ethanol) were painted evenly onto plastic film. After 24 hr, total number of eggs laid by a single mated female on control films (white bars) and BL-treated films (green bars) were counted. Data are mean ± SEM. For $10^{-5}$ M, $n = 9$, p=0.4364; for $10^{-4}$ M, $n = 12$, p=0.0275; for $10^{-3}$ M, $n = 13$, p=0.0015; $10^{-2}$ M, $n = 8$, p=0.0038. Data were analyzed using paired-samples *t*-test.

The online version of this article includes the following source data for figure 7:

**Source data 1.** Source data for *Figure 7*.

expression of defense-related genes (*Bari and Jones, 2009*). Jasmonic acid plays an important role in plant resistance to insects (*Wang et al., 2019*). Plants accumulate jasmonic acid and its derivatives upon wounding. Exogenous treatment with jasmonic acid activates the expression of hundreds of defense-related genes. Application of exogenous jasmonic acid to cabbage plants was shown to indirectly retard the development of *P. xylostella* larvae and reduce the pupal weight and female fecundity (*Lv and Liu, 2005*). Therefore, signaling molecules associated with induced plant defenses may be used as reliable cues by herbivorous insects. Up to now, only one study showed that *Helicoverpa zea* reacts to jasmonate and salicylate in plants, resulting in the activation of four of its cytochrome P450 genes that are associated with detoxification (*Li et al., 2002*). However, how the caterpillars eavesdrop the hormone signals remains a mystery. The present study provides the first evidence that *P. xylostella* can detect the plant hormone BL with a bitter GR. This reflects a new adaptation of insects to plant defenses.

Herbivorous insects have evolved counter adaptations against the chemical defenses of plants. The perception of bitter substances is an adaptation to avoid potentially toxic secondary plant metabolites. We can speculate that BL and EBL may influence insect development because BL and EBL have strikingly similar structures to that of ecdysteroid hormones in arthropods, such as 20-hydroxyecdysone (*Fujioka and Sakurai, 1997*). Their structures are so similar that BL and EBL show agonistic activity with 20-hydroxyecdysone in many insect species (*Zullo and Adam, 2002*). It has been shown that injection with 20 μg EBL was fatal to mid last-instar larvae of *Spodoptera littoralis* (*Smagghe et al., 2002*). The BL content differs widely among plant species; for example, it is $1.37 \times 10^{-4}$ g/kg in *Brassica campestris* L. leaves and $1.25 \times 10^{-6}$ g/kg in *Arabidopsis thaliana* leaves (*Lv et al., 2014*). Our results show that BL has the inhibitory effects on feeding and oviposition of *P. xylostella*, and the threshold concentration of BL for such behavioral inhibitions of *P.*

*xylostella* is in the range of $10^{-4}$–$10^{-3}$ g/kg, suggesting that BL plays a dual role of plant hormones and insect feeding/oviposition deterrents in plants.

There is a rich variety of bitter GRs in phytophagous insects, but only a few have been functionally characterized (*Kasubuchi et al., 2018*; *Zhang et al., 2019*). In this study, we showed that PxylGr34, a bitter GR highly expressed in larval head and adult antennae of *P. xylostella*, is tuned to the plant hormones BL and EBL, which mediates the aversive feeding/oviposition responses of *P. xylostella* to these compounds. These findings not only increase our understanding of the gustatory coding mechanisms of feeding/oviposition deterrents in phytophagous insects, but also offer new perspectives for using plant hormones as potential agents to suppress pest insects.

# Materials and methods

## Key resources table

| Reagent type (species) or resource | Designation | Source or reference | Identifiers | Additional information |
|---|---|---|---|---|
| Gene (*Plutella xylostella*) | Muscle actin gene | NCBI | GenBank: AB282645.1 | |
| Commercial assay or kit | RNeasy Plus Universal Mini Kit | Qiagen | Cat# 73404 | |
| Commercial assay or kit | Q5 High-Fidelity DNA Polymerase | NEB | Cat# M0491 | |
| Commercial assay or kit | M-MLV reverse transcriptase | Promega | Cat# M1701 | |
| Commercial assay or kit | SYBR Premix Ex TaqII | Takara | Cat# RR820 | |
| Commercial assay or kit | mMESSAGE mMACHINE SP6 | Ambion | Cat# AM1340 | |
| Chemical compound, drug | (+/−)-Abscisic acid | Sigma-Aldrich | CAS: 21293-29-8 | |
| Chemical compound, drug | L-Alanine | Sigma-Aldrich | CAS: 56-41-7 | |
| Chemical compound, drug | Allyl isothiocyanate | Sigma-Aldrich | CAS: 57-06-7 | |
| Chemical compound, drug | Aristolochic acid | Shanghai Macklin Biochemical Co., Ltd, China | CAS: 313-67-7 | |
| Chemical compound, drug | Brassinolide | Yuanyeshengwu Co., Ltd, China | CAS: 72962-43-7 | |
| Chemical compound, drug | (+/−)-Camphor | Beijing Mreda Technology Co., Ltd, China | CAS: 76-22-2 | |
| Chemical compound, drug | (−)-Citronellal | Tokyo Chemical Industry Co., Ltd, Japan | CAS: 5949-05-3 | |
| Chemical compound, drug | 24-Epibrassinolide | Shanghai Macklin Biochemical Co., Ltd, China | CAS: 78821-43-9 | |
| Chemical compound, drug | D-Fructose | Sigma-Aldrich | CAS: 7660-25-5 | |

*Continued on next page*

*Continued*

| Reagent type (species) or resource | Designation | Source or reference | Identifiers | Additional information |
|---|---|---|---|---|
| Chemical compound, drug | Gibberellic acid | Yuanyeshengwu Co., Ltd, China | CAS: 77-06-5 | |
| Chemical compound, drug | cis-3-Hexenyl salicylate | Shanghai Macklin Biochemical Co., Ltd, China | CAS: 65405-77-8 | |
| Chemical compound, drug | 20-Hydroxyecdysone | Yuanyeshengwu Co., Ltd, China | CAS: 5289-74-7 | |
| Chemical compound, drug | Indole-3-acetic acid | Yuanyeshengwu Co., Ltd, China | CAS: 87-51-4 | |
| Chemical compound, drug | (+/−)-Jasmonic acid | Tokyo Chemical Industry Co., Ltd, Japan | CAS: 6894-38-8 | |
| Chemical compound, drug | Kinetin | Sigma-Aldrich | CAS: 525-79-1 | |
| Chemical compound, drug | Methyl jasmonate | Sigma-Aldrich | CAS: 1211-29-6 | |
| Chemical compound, drug | Methyl salicylate | Sigma-Aldrich | CAS: 119-36-8 | |
| Chemical compound, drug | Myo-inositol | Sigma-Aldrich | CAS: 87-89-8 | |
| Chemical compound, drug | (+/−)-Nicotine | Sigma-Aldrich | CAS: 22083-74-5 | |
| Chemical compound, drug | Putrescine | Alfa Aesar | CAS: 110-60-1 | |
| Chemical compound, drug | Sinigrin | Sigma-Aldrich | CAS: 3952-98-5 | |
| Chemical compound, drug | Spermidine | Sigma-Aldrich | CAS: 124-20-9 | |
| Chemical compound, drug | Spermine | Sigma-Aldrich | CAS: 71-44-3 | |
| Chemical compound, drug | D-Sucrose | Sigma-Aldrich | CAS: 57-50-1 | |
| Software, algorithm | GraphPad Prism | GraphPad Prism | RRID:SCR_002798 | 8.3 |
| Software, algorithm | Adobe Illustrator | Adobe systems | RRID:SCR_014198 | CC2018 |
| Software, algorithm | pCLAMP software | pCLAMP software | RRID:SCR_011323 | |

## Insects and plants

*P. xylostella* was originally collected from the cabbage field of Institute of Plant Protection, Shanxi Academy of Agricultural Sciences, China. It is not specialized on peas. The insects were reared at the Institute of Zoology, Chinese Academy of Sciences, Beijing. The larvae were fed only with

cabbage (*Brassica oleracea* L.) and kept at 26 ± 1°C with a 16L:8D photoperiod and 55–65% relative humidity. The diet for adults was a 10% (v/v) honey solution. Pea (*Pisum sativum* L.) plants were grown in an artificial climate chamber at 26 ± 1°C with a 16L:8D photoperiod and 55–65% relative humidity. The plants were grown in nutrient soil in pots (8 × 8 × 10 cm) and were 4–5 weeks old when they were used in experiments.

### Care and use of *X. laevis*

All procedures were approved by the Animal Care and Use Committee of the Institute of Zoology, Chinese Academy of Sciences, and followed The Guidelines for the Care and Use of Laboratory Animals (protocol number IOZ17090-A). Female *X. laevis* were provided by Prof. Qing-Hua Tao (MOE Key Laboratory of Protein Sciences, Tsinghua University, China) and reared in our laboratory with pig liver as food. Six healthy naive *X. laevis* 18–24 months of age were used in these experiments. They were group-housed in a box with purified water at 20 ± 1°C. Before experiments, each *X. laevis* individual was anesthetized by bathing in an ice–water mixture for 30 min before surgically collecting the oocytes.

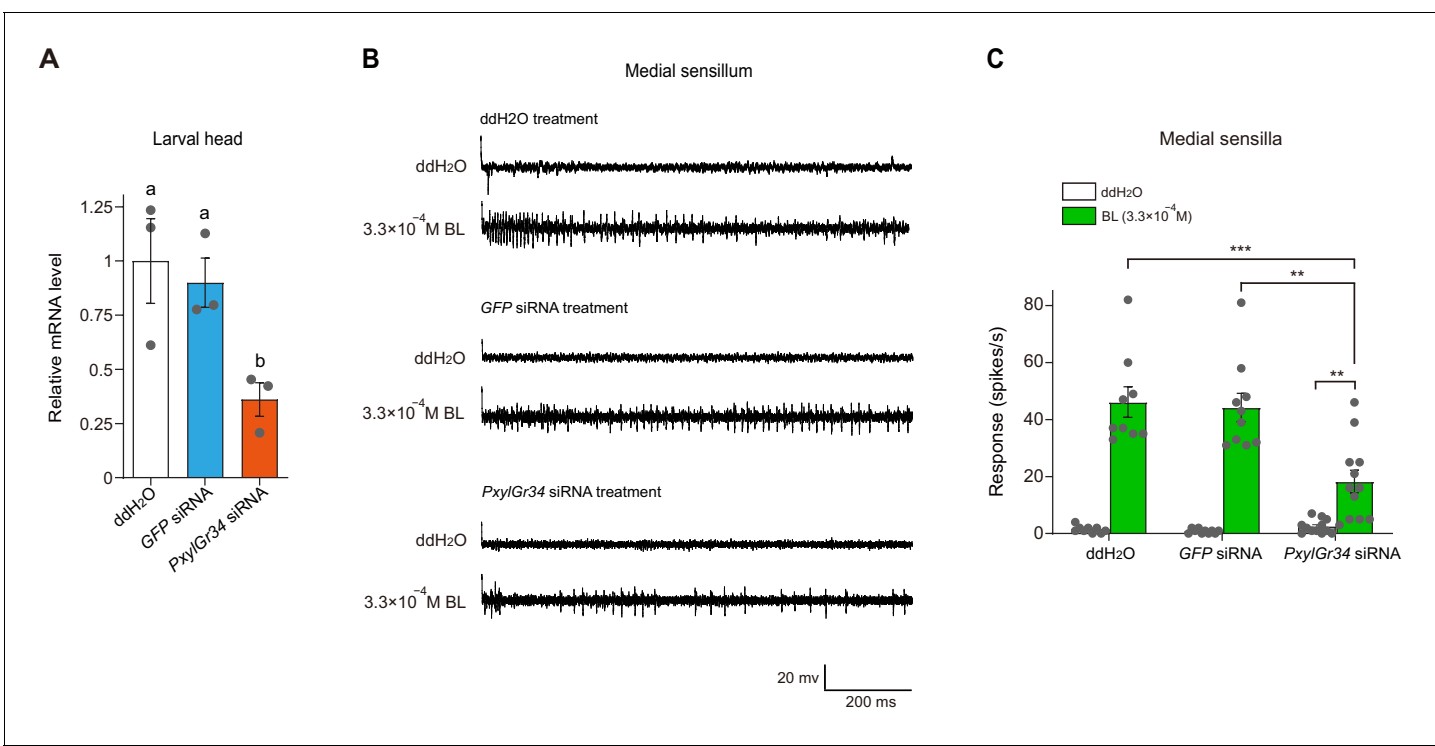

**Figure 8.** PxylGr34 siRNA-treated *P. xylostella* larvae inhibit the responses of sensilla to brassinolide (BL). (**A**) Effect of PxylGr34 siRNA on transcript levels of *PxylGr34* in 4th instar larval head. Heads were collected after feeding larvae with cabbage leaf discs coated with *PxylGr34* siRNA, *GFP* siRNA, or ddH$_2$O. Relative transcript levels of *PxylGr34* in each treatment were determined by qPCR. Data are mean ± SEM. n = 3 replicates of 21–24 heads each, p=0.0237 (one-way ANOVA, Tukey's HSD test). (**B**) Typical electrophysiological recordings in response to water and BL at 3.3 × 10$^{-4}$ M for 1 s obtained by tip-recording from a neuron innervating the medial sensillum styloconica, on maxillary galea of *P. xylostella* 4th instar larvae with cabbage leaf discs coated with *PxylGr34* siRNA, *GFP* siRNA, or ddH$_2$O. (**C**) Response profiles of the medial sensilla styloconica on the maxilla of *P. xylostella* 4th instar larvae with cabbage leaf discs coated with *PxylGr34* siRNA, *GFP* siRNA, or ddH$_2$O, to water and BL at 3.3 × 10$^{-4}$ M. Data are mean ± SEM. The data of each treatment to water and BL at 3.3 × 10$^{-4}$ M were analyzed by paired-samples *t*-test. For ddH$_2$O treatment, n = 9, p<0.0001; for *GFP* siRNA treatment, n = 10, p<0.0001; for *PxylGr34* siRNA treatment, n = 12, p=0.0019. The data of different treatments to BL at 3.3 × 10$^{-4}$ M were analyzed by one-way ANOVA, Tukey's HSD test, p=0.0002.

The online version of this article includes the following source data for figure 8:

**Source data 1.** Source data for *Figure 8A and C*.

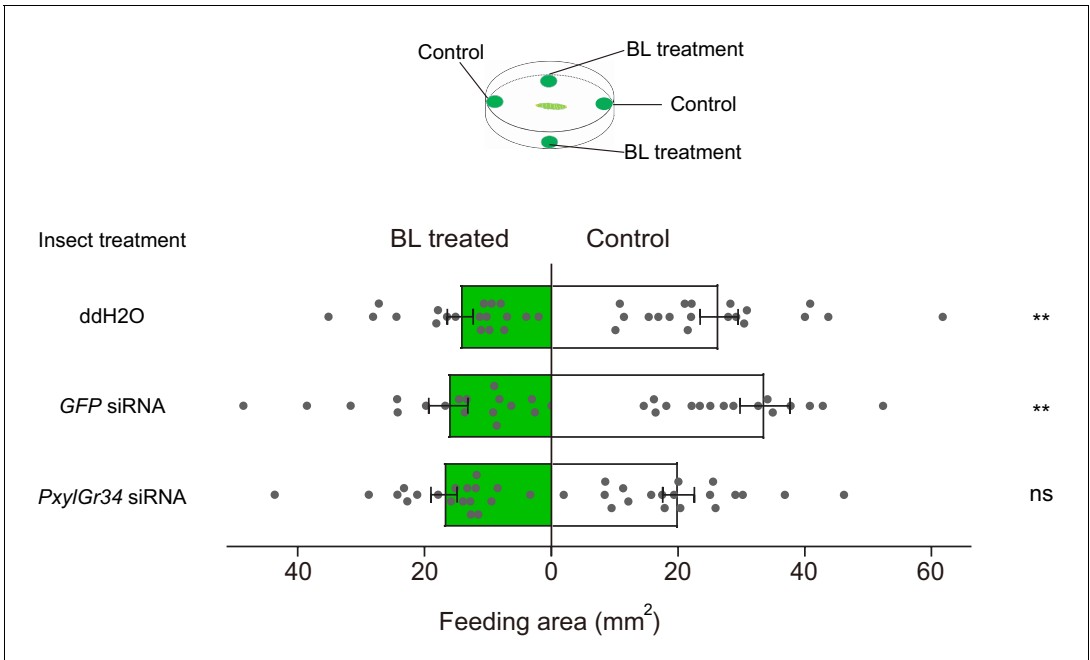

**Figure 9.** PxylGr34 siRNA-treated *P. xylostella* larvae alleviated the feeding deterrent effect of brassinolide (BL). Choice assay using 4th instar larvae fed on cabbage leaf discs treated with *PxylGr34* siRNA, *GFP* siRNA, or ddH2O. In dual-choice assay, larvae chose between control pea leaf discs (treated with 50% ethanol) and those treated with BL at $10^{-4}$ M (diluted in 50% ethanol). Figure shows total feeding area of control (white bars) and treated discs (green bars). Data are mean ± SEM. For ddH2O treatment, n = 19, p=0.0093; for *GFP* siRNA treatment, n = 18, p=0.0044; for *PxylGr34* siRNA treatment, n = 19, p=0.1864. Data were analyzed by paired-samples *t*-test.

The online version of this article includes the following source data for figure 9:

**Source data 1.** Source data for *Figure 9*.

## Sequencing and expression analysis of GR genes in *P. xylostella*

We conducted transcriptome analyses of the *P. xylostella* moth antennae, foreleg (only tibia and tarsi), head (without antenna), and the 4th instar larval mouthparts. Total RNA was extracted using QIAzol Lysis Reagent (Qiagen, Hilden, Germany) and treated with DNase I following the manufacturer's protocol. Poly(A) mRNA was isolated using oligo dT beads. First-strand complementary DNA was generated using random hexamer-primed reverse transcription, followed by synthesis of second-strand cDNA using RNaseH and DNA polymerase I. Paired-end RNA-seq libraries were prepared following Illumina's protocols and sequenced on the Illumina HiSeq 2500 platform (Illumina, San Diego, CA). High-quality clean reads were obtained by removing adaptors and low-quality reads, then de novo assembled using the software package Trinity v2.8.5 (*Haas et al., 2013*). The GR genes were annotated by NCBI BLASTX searches against a pooled insect GR database, including GRs from *P. xylostella* (*Engsontia et al., 2014*; *You et al., 2013*; *Yang et al., 2017*) and other insect species (*Guo et al., 2017*; *Xu et al., 2016*; *Robertson et al., 2003*). The translated amino acid sequences of the identified GRs were aligned manually by NCBI BLASTP and tools at the T-Coffee web server (*Notredame et al., 2000*). The TPM values were calculated using the software package RSEM v1.2.28 (*Li and Dewey, 2011*) to analyze GR gene transcript levels.

## Phylogenetic analysis

Phylogenetic analysis of *P. xylostella* GRs was performed based on amino acid sequences, together with those of previously reported GRs of *Heliconius melpomene* (*Briscoe et al., 2013*) and *B. mori* (*Guo et al., 2017*). Amino acid sequences were aligned with MAFFT v7.455 (*Rozewicki et al., 2019*), and gap sites were removed with trimAl v1.4 (*Capella-Gutiérrez et al., 2009*). The maximum likelihood phylogenetic tree was constructed using RAxML v8.2.12 (*Stamatakis, 2014*) with the Jones-Taylor-Thornton amino acid substitution model. Node support was assessed using a bootstrap

method based on 1000 replicates. The tree was visualized in FigTree Version 1.4.4 (http://tree.bio.ed.ac.uk/software/figtree/).

## RNA isolation and cDNA synthesis

The adult antennae, head (without antennae), forelegs (only tarsi and tibia), and ovipositor, and the larval head, thorax (without wing disc, thoracic legs, gut, or other internal tissues), thoracic leg, abdomen (without gut or other internal tissues) and gut were dissected immediately placed in a 1.5 mL Eppendorf tube containing liquid nitrogen, and stored at −80°C until use. Total RNA was extracted using QIAzol Lysis Reagent following the manufacturer's protocol (including DNase I treatment), and RNA quality was checked with a spectrophotometer (NanoDrop 2000; Thermo Fisher Scientific, Waltham, MA, USA). The single-stranded cDNA templates were synthesized using 2 μg total RNAs from various samples with 1 μg oligo (dT) 15 primer (Promega, Madison, WI, USA). The mixture was heated to 70°C for 5 min to melt the secondary structure of the template, then M-MLV reverse transcriptase (Promega) was added and the mixture was incubated at 42°C for 1 hr. The products were stored at −20°C until use.

## PCR amplification of *PxylGr34* from *P. xylostella*

Based on the candidate full-length nucleotide sequences of *PxylGr34* identified from our transcriptome data, we designed specific primers (*Supplementary file 1*). All amplification reactions were performed using Q5 High-Fidelity DNA Polymerase (New England Biolabs, Beverly, MA, USA). The PCR conditions for amplification of *PxylGr34* were as follows: 98°C for 30 s, followed by 30 cycles of 98°C for 10 s, 60°C for 30 s, and 72°C for 1 min, and final extension at 72°C for 2 min. Templates were obtained from antennae of female *P. xylostella*. The sequences were further verified by Sanger sequencing.

## Quantitative real-time PCR

The qPCR analyses were conducted using the QuantStudio 3 Real-Time PCR System (Thermo Fisher Scientific) with SYBR Premix Ex *Taq* (TaKaRa, Shiga, Japan). The gene-specific primers to amplify an 80–150 bp product were designed by Primer-BLAST (http://www.ncbi.nlm.nih.gov/tools/primer-blast/) (*Supplementary file 1*). The thermal cycling conditions were as follows: 10 s at 95°C, followed by 40 cycles of 95°C for 5 s and 60°C for 31 s, followed by a melting curve analysis (55–95°C) to detect a single gene-specific peak and confirm the absence of primer dimers. The product was verified by nucleotide sequencing. *PxylActin* (GenBank number: AB282645.1) was used as the control gene (*Teng et al., 2012*). Each reaction was run in triplicate (technical replicates) and the means and standard errors were obtained from three biological replicates. The relative copy numbers of *PxylGr34* were calculated using the $2^{-\Delta\Delta Ct}$ method (*Livak and Schmittgen, 2001*).

## Receptor functional analysis

The full-length coding sequence of PxylGr34 was first cloned into the pGEM-T vector (Promega) and then subcloned into the pCS2$^+$ vector. cRNA was synthesized from the linearized modified pCS2$^+$ vector with mMESSAGE mMACHINE SP6 (Ambion, Austin, TX, USA). Mature healthy oocytes were treated with 2 mg mL$^{-1}$ collagenase type I (Sigma-Aldrich, St Louis, MO, USA) in Ca$^{2+}$-free saline solution (82.5 mM NaCl, 2 mM KCl, 1 mM MgCl$_2$, 5 mM HEPES, pH = 7.5) for 20 min at room temperature. Oocytes were later microinjected with 55.2 ng cRNA. Distilled water was microinjected into oocytes as the negative control. Injected oocytes were incubated for 3–5 days at 16°C in a bath solution (96 mM NaCl, 2 mM KCl, 1 mM MgCl$_2$, 1.8 mM CaCl$_2$, 5 mM HEPES, pH = 7.5) supplemented with 100 mg mL$^{-1}$ gentamycin and 550 mg mL$^{-1}$ sodium pyruvate. Whole-cell currents were recorded with a two-electrode voltage clamp. The intracellular glass electrodes were filled with 3 M KCl and had resistances of 0.2–2.0 MΩ. Signals were amplified with an OC-725C amplifier (Warner Instruments, Hamden, CT, USA) at a holding potential of −80 mV, low-pass filtered at 50 Hz, and digitized at 1 kHz. Each of 24 compounds listed in Key Resources Table was diluted and the pH was adjusted to 7.5 in Ringer's solution before being introduced to the oocyte recording chamber using a perfusion system. Data were acquired and analyzed using Digidata 1322A and pCLAMP software (RRID:SCR_011323) (Axon Instruments Inc, Foster City, CA, USA). Dose-response data were analyzed using GraphPad Prism software (RRID:SCR_002798 6) (GraphPad Software Inc, San Diego, CA, USA).

## Electrophysiological responses of contact chemosensilla on the maxilla of larvae to BL, EBL, and sinigrin

The tip-recording technique was used to record action potentials from the lateral and medial sensilla styloconica on the maxillary galea of larvae, following the protocols described elsewhere (*Hodgson et al., 1955*; *van Loon, 1990*; *van Loon et al., 2002*). Distilled water served as the control stimulus, as KCl solutions elicited considerable responses from the galeal styloconic taste sensilla (*van Loon et al., 2002*), and sinigrin served as the positive control stimulus (*van Loon et al., 2002*). Experiments were carried out with larvae that were 1–2 days into their final stadium (4th instar). The larvae were starved for 15 min before analysis. The larvae were cut at the mesothorax, and then silver wire was placed in contact with the insect tissue. The wire was connected to a preamplifier with a copper miniconnector. A glass capillary filled with the test compound, into which a silver wire was inserted, was placed in contact with the sensilla. Electrophysiological responses were quantified by counting the number of spikes in the first second after the start of stimulation. The interval between two successive stimulations was at least 3 min to avoid adaptation of the tested sensilla. Before each stimulation, a piece of filter paper was used to absorb the solution from the tip of the glass capillary containing the stimulus solution to avoid an increase in concentration due to evaporation of water from the capillary tip. The temperature during recording ranged from 22° to 25°C. Neural activity was sampled with a computer equipped with a Metrabyte DAS16 A/D conversion board. An interface was used (GO-box) for signal conditioning. This involved a second order band pass filter ($-3$ dB frequencies: 180 and 1700 Hz) (*van Loon et al., 2002*). The electrophysiological signals were recorded by SAPID Tools software version 16.0 (*Smith et al., 1990*), and analyzed using Autospike software version 3.7 (Syntech).

## Dual-choice feeding bioassays

Dual-choice feeding assays were used to quantify the behavioral responses of *P. xylostella* larvae to BL and EBL on pea leaves. Pea (*Pisum sativum* L.) is a neutral host plant of *P. xylostella* and widely used in the feeding preference analysis in this species (*Thorsteinson, 1953*; *van Loon et al., 2002*). These assays were based on the protocol reported by *van Loon et al., 2002*, with modifications. The axisymmetric pinnate leaf was freshly picked from 4-week-old pea plants grown in a climate-controlled room. One leaf was folded in half, and two leaf discs (diameter, 7 mm) were punched from the two halves as the control (C) and treated (T) discs, respectively. For the treated discs, 5 μL (13 μL/cm$^2$) of the test compound diluted in 50% ethanol was spread on the upper surface using a paint brush. For the control discs, 5 μL 50% ethanol was applied in the same way. Control (C) and treated (T) discs were placed in a C-T-C-T sequence around the circumference of the culture dish (60 mm diameter ×15 mm depth; Corning, NY, USA). After the ethanol had evaporated (15 min later), a single 4th instar caterpillar (day 1), which had been starved for 6 hr, was placed in each dish. The dishes were kept for 24 hr at 23°–25°C in the dark, to avoid visual stimuli. Each dish was covered with a circular filter paper disc (diameter 7 cm) moistened with 200 μL ddH$_2$O to maintain humidity. At the end of the test, the leaf discs were scanned using a DR-F120 scanner (Canon, Tokyo, Japan) and the remaining leaf area was quantified with ImageJ software (NIH). Paired-sample *t*-test was used to detect differences in the consumed leaf area between control and treated leaf discs.

## Dual-choice oviposition bioassays

A dual-choice oviposition bioassay was used to quantify the behavioral responses of *P. xylostella* mated female moths to BL. This assay was modified from the protocol reported by *Gupta and Thorsteinson, 1960* and *Justus and Mitchell, 1996*. A paper cup (10 cm diameter ×8 cm height) with a transparent plastic lid (with 36 pinholes for ventilation) was used for ovipositing of the mated females. Fresh cabbage leaf juice was centrifuged at 3000 rpm for 5 min, and 60 μL of the supernatant was spread with a paintbrush onto polyethylene (PE) film from clinical gloves for 10 min to evaporate. Four culture dishes (35 mm diameter) (Corning, New York, NY, USA) covered with these PE films were placed on the bottom of each cup. This oviposition system was developed based on the biology of *P. xylostella* (*Harcourt, 1957*). On each of two diagonally positioned treatment films, 125 μL BL (13 μL/ cm$^2$) diluted in 50% ethanol was evenly spread on the upper surface using a paint brush. On the other two diagonally positioned films, 125 μL 50% ethanol (control) was spread in the

same way. After the ethanol had evaporated (30 min later), a small piece of absorbent cotton soaked with 10% honey-water mixture was placed in the center of the cup.

The pupae of *P. xylostella* were selected and newly emerged adults were checked and placed in a mesh cage (25 × 25 × 25 cm), with a 10% honey-water mixture supplied during the light phase. The female:male ratio was 1:3 to ensure that all the females would be mated. After at least 24 hr of mating time, the mated females were removed from the cage and placed individually into the oviposition cup during the light phase. After 24 hr at 26 ± 1°C with a 16L: 8D photoperiod and 55–65% relative humidity, the number of eggs on each plastic film was counted. Paired-samples *t*-test was used to detect significant differences in the number of eggs laid between the control and treatment films. siRNA preparation.

A unique siRNA region specific to *PxylGr34* was selected guided by the siRNA Design Methods and Protocols (*Takasaki, 2013*). The siRNA was prepared using the T7 RiboMAX Express RNAi System kit (Promega, Madison) following manufacturer's protocol. The GFP (GenBank: M62653.1) siRNA was designed and synthesized using the same methods. We tested three different siRNAs of *PxylGr34* based on different sequence regions, and selected the most effective and stable one for further analyses. The oligonucleotides used to prepare siRNAs are listed in *Supplementary file 2*.

### Oral delivery of siRNA

The siRNAs were supplied to the larvae by oral delivery as reported elsewhere (*Chaitanya et al., 2017*; *Gong et al., 2011*), with some modifications. Each siRNA was spread onto cabbage leaf discs (*Brassica oleracea*) and fed to 4th-instar larvae. Freshly punched cabbage leaf discs (diameter 0.7 cm) were placed into 24-well clear multiple well plates (Corning, NY, USA). For each disc, 3 µg siRNA diluted in 6 µL 50% ethanol was evenly distributed on the upper surface using a paint brush. Both 50% ethanol and GFP siRNA were used as negative controls. After the ethanol had evaporated (20 min later), one freshly molted 4th-instar larva, which had been starved for 6 hr, was carefully transferred onto each disc and then allowed to feed for 12 hr. Each well was covered with dry tissue paper to maintain humidity. The larvae that had consumed the entire disc were selected and starved for another 6 hr, and then these larvae were used in the dual-choice behavioral assay, electrophysiological responses of contact chemosensilla on the maxilla, or for qPCR analyses as described above. The larvae that did not consume the treated discs were discarded.

### Data analysis

Data were analyzed using GraphPad Prism 8.3. Figures were created using GraphPad Prism 8.3 and Adobe illustrator CC 2018 (Adobe systems, San Jose, CA). Two-electrode voltage-clamp recordings, electrophysiological dose-response curves, and the square-root transformed qPCR data were analyzed by one-way ANOVA and Tukey's HSD tests with two distribution tails. These analyses were performed using GraphPad prism 8.3. Electrophysiological response data and all dual-choice test data were analyzed using the two-tailed paired-samples *t*-test. Statistical tests and the numbers of replicates are provided in the figure legends. In all statistical analyses, differences were considered significant at $p < 0.05$. Asterisks represent significance: *$p < 0.05$, **$p < 0.01$, ***$p < 0.001$, ****$p < 0.0001$; ns, not significant. Response values are indicated as mean ± SEM; and *n* represents the number of samples in all cases.

## Acknowledgements

We thank the lab members Hao Guo, Jun Yang, Nan-Ji Jiang, and Rui Tang for their helps in the data analysis and comments, Yan Chen, Zi-Lin Li, Shuai-Shuai Zhang, and Ruo-Xi Shi for their assistance in tip-recording analysis. We thank Xi-Zhong Yan from Shanxi Agricultural University for the assistance in insect rearing, Prof. Qing-Hua Tao from MOE Key Laboratory of Protein Sciences, Tsinghua University for providing *Xenopus laevis* frogs. We also thank Prof. Bill Hansson from Max Planck Institute for Chemical Ecology, Germany, for his comments on this work. This work is funded by National Key R and D Program of China (Grant No. 2017YFD0200400), the National Natural Science Foundation of China (Grant No. 31830088), and China Postdoctoral Science Foundation (Grant No. 2019M660792).

## Additional information

### Funding

| Funder | Grant reference number | Author |
| --- | --- | --- |
| National Natural Science Foundation of China | 31830088 | Chen-Zhu Wang |
| China Postdoctoral Science Foundation | 2019M660792 | Ke Yang |
| National Key R and D Program of China | 2017YFD0200400 | Chen-Zhu Wang |

The funders had no role in study design, data collection and interpretation, or the decision to submit the work for publication.

### Author contributions

Ke Yang, Conceptualization, Resources, Data curation, Software, Formal analysis, Funding acquisition, Validation, Investigation, Visualization, Methodology, Writing - original draft, Project administration, Writing - review and editing, Experiment design; Xin-Lin Gong, Validation, Investigation; Guo-Cheng Li, Chao Ning, Formal analysis, Validation; Ling-Qiao Huang, Resources, Validation, Investigation; Chen-Zhu Wang, Conceptualization, Resources, Data curation, Software, Formal analysis, Supervision, Funding acquisition, Validation, Investigation, Visualization, Methodology, Writing - original draft, Project administration, Writing - review and editing, Experiment design

### Author ORCIDs

Ke Yang (iD) https://orcid.org/0000-0002-4138-3373
Chen-Zhu Wang (iD) https://orcid.org/0000-0003-0418-8621

### Ethics

Animal experimentation: All procedures were approved by the Animal Care and Use Committee of the Institute of Zoology, Chinese Academy of Sciences, and followed The Guidelines for the Care and Use of Laboratory Animals (protocol number: IOZ17090-A).

### Decision letter and Author response

Decision letter https://doi.org/10.7554/eLife.64114.sa1
Author response https://doi.org/10.7554/eLife.64114.sa2

## Additional files

### Supplementary files

• Supplementary file 1. Sequence information for gustatory receptors of *Plutella xylostella*.

• Supplementary file 2. Primers used for qPCR, *Xenopus* oocyte expression (Xe), and siRNA synthesis.

• Transparent reporting form

### Data availability

All data generated or analysed during this study are included in the manuscript and supporting files. Source data files have been provided for all the figures.

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
