## [Decision Letter]

**Acceptance summary:**

The vast majority of herbivorous insects are specialized on a set of host plants and use a combination of lineage-specific and more ubiquitous chemicals synthesized by plants to discriminate among potential hosts. This study identifies a gustatory receptor in a mustard-specialized moth that responds to the plant steroid brassinide. This is an important advance as very few Grs have been implicated in the role of detecting plant compounds among truly herbivorous insects.

**Decision letter after peer review:**

[Editors’ note: the authors submitted for reconsideration following the decision after peer review. What follows is the decision letter after the first round of review.]

Thank you for submitting your work entitled "A gustatory receptor tuned to the steroid plant hormone brassinolide in *Plutella xylostella* (Lepidoptera: Plutellidae)" for consideration by *eLife*. Your article has been reviewed by three peer reviewers, and the evaluation has been overseen by a Reviewing Editor and a Senior Editor. The following individual involved in review of your submission has agreed to reveal their identity: Noah Whiteman (Reviewer #1).

Our decision has been reached after consultation between the reviewers. Based on these discussions and the individual reviews below, we regret to inform you that your work will not be considered further for publication in *eLife* in its current form. Please note that *eLife* aims for an efficient peer review process and rejects submissions in cases where revisions are likely to take longer than two months. However, should you address the concerns summarized in the following paragraph, we would be happy to consider a revised study as a new submission.

As you will see from the reviews below, the reviewers thought that the topic is interesting but raised conceptual and technical concerns which would require additional experiments to fully address. Reviewers 1 and 2 raised concerns about the framing of the study that require addressing. Reviewer 3 raised concerns about the tip recordings and behavioral experiments that also require addressing. The reviewers agree that it is essential for the authors to perform additional SSR experiments including knockdown animals and e.g. sinigrin as a positive control. The reviewers also recommend that the authors revise the text and figures, including some reorganization (e.g. move EBL experiments to supplementals). Reviewer three noted that a more detailed molecular characterization of PxylGr34 would be interesting, but the reviewers and editors agree that this would be beyond the scope of the study and do not consider it to be an essential revision.

Reviewer #1:

The vast majority of herbivorous insects are specialized on a set of host plants and use a combination of lineage-specific and more ubiquitous chemicals synthesized by plants to discriminate among potential hosts. Mechanistic information on this question is lacking, specifically regarding the identify and evolution of chemoreceptors that are responsible for mediating these behaviors, which are central to understanding the evolution of herbivorous insects. Although progress has been made on non-herbivorous drosophilids and two herbivorous drosophilids, we still lack basic mechanistic information from most herbivorous lineages. Here, Yang et al. focus on this question using the mustard-specialized diamondback moth *Plutella xylostella*. Through RNA sequencing and qPCR, they identified PxylGr34 as being highly expressed in adult female antennae and in larval tissues. By expressing PxylGr34 in *Xenopus oocytes* and subjecting them to 24 diverse phytochemicals, they found that the canonical plant hormones brassinolide and 24-epibrassinolide induce a strong response at increasing concentrations. This electrophysiological response is similarly seen from recordings from larval medial (but not lateral) sensilla styloconica. Behavioral assays with larvae and mated adult females in which individuals are given a choice between pea discs/plastic film, respectively, treated and untreated with BL, show aversion to BL in both life stages. In larvae, this aversion is reduced by feeding them PxylGr34 siRNA, which significantly, but not completely, reduced mRNA expression of PxylGr34 in their heads. Collectively, these results show compelling evidence that PxylGr34 very likely mediates the response to BL. While there is clearly still much to learn in terms of understanding the coding mechanisms of Grs, i.e. whether other Grs are necessary for this BL-induced response, this is significant because very few Grs have been implicated in the role of detecting plant compounds among truly herbivorous insects. This is also the first time a chemoreceptor has been identified that seems to be tuned to this particular plant hormone, a ubiquitous plant steroid. Where this manuscript can improve is the general framework and discussion and the importance of discriminating diverse phytochemicals and in particular how insects can use host cues associated with stress, including plant hormones, to avoid particular plants.

1) The expression patterns of PxylGr34 is really striking, which supports their interest in focusing on it. I think the framing of the paper isn't doing it justice. In the Results section under "Identification of the GR gene PxylGr34 in *P. xylostella*" they would benefit from framing it in a question driven way, i.e. they were interested in searching for candidate Grs that may be involved in host selection within a model herbivore, and they looked for candidates among those that had been annotated from the *P. xylostella* genome. They then looked specifically for candidates that were highly expressed, comparing several chemosensory organs. Intriguingly, PxylGr34 was highly expressed in the antennae, as well as in other gustatory tissues, relative to other Grs. Because the expression data across Grs provides the biggest piece of evidence for focusing on this Gr, I would pull Figure 1—figure supplement 2 out from the supplement and stick it in Figure 1 as an additional panel. That being said, just because it is highly expressed, doesn't mean that it is the most important Gr for enabling feeding discrimination on their host plants – it would certainly be interesting to do the same experiments with all the other Grs but that's out of the scope of this paper for now.

2) "Brassinolide (BL) is a C28 brassinosteroid (BR)…" this whole paragraph needs to be restructured so that it's clearer why they focused on brassinolides, a plant hormone, rather than the typical anti-herbivore plant defenses, as mentioned in the previous sentence, or rather than other plant hormones. They outline some interesting reasons, i.e. their ubiquity in the plant kingdom, their structural similarity to molting hormones, but more explanation would benefit their study-there is still a way to frame this that makes the choice to focus on BLs clearer. After reading the rest of the paper, it's clear now that they tested 24 different phytochemicals of various groups, and brassinolide was the one that activated PxylGr34. But it makes sense for the Introduction to discuss more generally what their initial thoughts or hypotheses were about the kinds of compounds that should be important for herbivorous insects. If they had an a priori reason to be interested in brassinolides, then discuss that here; otherwise, leave it for the Discussion section.

3) Subsection “BL and its analog EBL induced a strong response in the oocytes expressing PxylGr34” – it would be great to mention earlier in this paper that 24 phytochemicals were tested, in the Abstract and Introduction. It would have answered my earlier questions about why there was a focus on brassinolides. Also, it might be worth mentioning that brassinolides might not be the only stimulant of Gr34; although 24 is a rather large number of compounds to test, there still may be others. Also, I only see the chemical listed in Figure 2. It would be great to list them at least in the Materials and methods, with source information.

4) Discussion paragraph six – it would make sense if this last paragraph or some part of the discussion was framed around why insects would want to detect brassinolides. That's the main question I've had throughout this paper. Is there evidence that insects routinely detect plant hormones to regulate their behavior (I know this to be true, but a paragraph on this topic csould be important)? Do these hormones correlate well with the nutritional content, growth, stress, defense concentrations, etc. within the plants to provide a reliable cue for where to feed? From there, you can then speculate that perhaps the BL may influence insect development because of the structural similarity with ecdysteroids, and then that discussion flows well from there.

5) Subsection “Phylogenetic analysis” – neighbor joining (NJ) method – NJ is not a true phylogenetic method per se (it ignores character information completely), and so I suggest you repeated this analysis by using maximum likelihood inference or Bayesian inference (BI). NJ is based on genetic distances, whereas ML or BI uses the character information and a model of sequence evolution to explore the tree space before selecting the most likely tree. The general consensus is that more robust phylogenetic methods should be used for datasets like these.

6) Figure 1 and Figure 3: For Figure 1, I'd like to see all datapoints overlaid on each histogram. Same with Figure 3B and 3D and Figure 4, 5, 6 where practicable. All of the raw data should be deposited in an excel sheet that is easily accessible, so we can link each figure to a dataset readily (data for each figure on a different sheet in same file).

Reviewer #2:

The manuscript describes the expression pattern of gustatory receptors in taste organs of larvae and females of the cabbage moth. One of these receptors (GR34) belonging to the bitter receptors was found to have an especially high expression level. Heterologous expression of GR34 in *Xenopus oocytes* revealed a ubiquitous plant hormone, brassinolide, as the best ligand. Brassinolide turned out to be deterrent for foraging larvae and egg laying females. After silencing the expression of GR34 in larvae, the deterrent effect of the plant hormone was abolished. Although the manuscript adds valuable information about the function of gustatory receptors in insects, several points are not clearly enough explained to reach a broader audience.

1) Novelty of the research: The authors stress in the Introduction that the molecular basis of deterrent gustatory neurons in phytophagous insects has not been investigated yet. However, one publication they cite at the end of their discussion (Kasubuchi et al., 2018) could deorphanize several bitter GRs in the silk moth, and show the deterrent effect of a ligand of these GRs in larvae. It would be reasonable to mention this study already in the Introduction, and state in which way the present manuscript wants to add further insights, e.g. by silencing the receptor and testing oviposition behavior.

2) Role of the plant hormone brassinolide: It should be explained why it is interesting to test if insects can detect this ubiquitous plant hormone. Is brassinolide regularly tested in insect gustatory research? What kind of information might high doses of brassinolide that are necessary to elicit physiological and behavioral responses convey for the hungry larvae or egg-laying female? As the hormone is present across the plant kingdom and in almost every plant tissue it is difficult to imagine how it could be used to identify host plants.

3) Molecular work: A) How were the 21 GRs analyzed in Figure 1—figure supplement 2 selected from the 42 bitter GRs identified the phylogenetic tree? B) Why is there a difference in the expression pattern of GR34 in female tissue in Figure 1-S3 (antenna, head and foreleg) and in Figure 1 (only antenna)?

4) Effect of GR34 in egg laying behavior: The focus of the study was on larvae; however, it would be worth exploring the role of GR34 in female moths. Would it be possible to record from antennal gustatory sensilla of the female, and to silence GR34 also in females and investigate the effect on oviposition? It should be mentioned in the Introduction or Discussion that Plutella females touch the leaf surfaces with their antenna before egg laying to explain why gustatory receptors on the antenna might be useful for host plant choice.

Reviewer #3:

In this manuscript, the authors characterized PxylGr34 of *P. xylostella*. They identified the full-length coding sequencing of PxylGr34 based on public data and their unpublished RNA-seq data. They found that PxylGr34 is highly expressed in larva head as well as female antennae in a real-time PCR. Through functional studies, they show that PxylGr34 is specifically tuned to BL and EBL in *Xenopus* oocyte in vitro system. They further show that BL evoked electrophysiological response in the medial sensilla styloconica on maxillary galea of larvae. Finally, they show BL inhibited larvae feeding and female oviposition of *P. xylostella*.

The authors have done a good job of characterizing PxylGr34 in vitro analysis. However, this manuscript has to remedy several limitations to be published in *eLife*.

1) It will be great if they can show the molecular properties of PxylGr34- for example, whether they are ligand-gated channels or G-protein coupled receptors.

2) There are several issues with tip-recording.

i) It is not clear whether PxylGr34 is required for BL-evoked in vivo electrophysiological responses. A knockdown experiment is recommended.

i) Representative traces are hard to read, making it difficult to determine whether these are actual spikes upon BL stimulation. Also, please add a y-axis scale bar.

iii) Since it has been reported that Sinigrin and glucosinolates stimulate the same sensilla, it is beneficial to show the spikes evoked by these chemicals as a control.

iv) Please show whether EBL can also stimulate the GSNs in the medial sensilla.

v) It will be beneficial to show the schematic of larval taste organs.

3) The effect of EBL is not convincing. It will be helpful to provide a dose-response profile with a wide range of EBL.

4) It is hard to tell if the knockdown of PxylGr34 has an effect on feeding behavior. It seems that only control food feeding increased, in contrast to BL-containing food.

5) It is interesting whether PxylGr34 is only expressed in adult female antenna and involved in ovipositing. Is it expressed in male antenna? If it is, what is the role of PxylGr34 in male?

6) There is no description for Figure 2D in the main text.

7) The authors have to provide the experimental condition and data of the survival test with BL.

8) Typo Figure 1 legend “4th instar larvae” to “adult female.”

In general, this manuscript is premature and does not have a significant scientific impact to general readers of *eLife*.

---

## [Author Response]

[Editors’ note: the authors resubmitted a revised version of the paper for consideration. What follows is the authors’ response to the first round of review.]

Reviewer #1:The vast majority of herbivorous insects are specialized on a set of host plants and use a combination of lineage-specific and more ubiquitous chemicals synthesized by plants to discriminate among potential hosts. Mechanistic information on this question is lacking, specifically regarding the identify and evolution of chemoreceptors that are responsible for mediating these behaviors, which are central to understanding the evolution of herbivorous insects. Although progress has been made on non-herbivorous drosophilids and two herbivorous drosophilids, we still lack basic mechanistic information from most herbivorous lineages. Here, Yang et al. focus on this question using the mustard-specialized diamondback moth Plutella xylostella. Through RNA sequencing and qPCR, they identified PxylGr34 as being highly expressed in adult female antennae and in larval tissues. By expressing PxylGr34 in *Xenopus* oocytes and subjecting them to 24 diverse phytochemicals, they found that the canonical plant hormones brassinolide and 24-epibrassinolide induce a strong response at increasing concentrations. This electrophysiological response is similarly seen from recordings from larval medial (but not lateral) sensilla styloconica. Behavioral assays with larvae and mated adult females in which individuals are given a choice between pea discs/plastic film, respectively, treated and untreated with BL, show aversion to BL in both life stages. In larvae, this aversion is reduced by feeding them PxylGr34 siRNA, which significantly, but not completely, reduced mRNA expression of PxylGr34 in their heads. Collectively, these results show compelling evidence that PxylGr34 very likely mediates the response to BL. While there is clearly still much to learn in terms of understanding the coding mechanisms of Grs, i.e. whether other Grs are necessary for this BL-induced response, this is significant because very few Grs have been implicated in the role of detecting plant compounds among truly herbivorous insects. This is also the first time a chemoreceptor has been identified that seems to be tuned to this particular plant hormone, a ubiquitous plant steroid. Where this manuscript can improve is the general framework and discussion and the importance of discriminating diverse phytochemicals and in particular how insects can use host cues associated with stress, including plant hormones, to avoid particular plants.

Thank you very much for your positive comments on our study. The summarization is very thoughtful and constructive. We agree that there is clearly still much to learn in terms of understanding the coding mechanisms of GRs. Based on our results, single GR (PxylGr34) expression is enough to cause *Xenopus oocytes* to respond to BL, implying other Grs are not necessary for this BL-induced response, but there is a possibility that this GR coupled with other GRs may have some new functions, which is what we would focus on in the next. In the new version, we revise the general framework, and rewrite two paragraphs in the Discussion about the insects using phytohormones associated with induced plant defenses as reliable cues, and their counter adaptations against the chemical defenses of plants.

1) The expression patterns of PxylGr34 is really striking, which supports their interest in focusing on it. I think the framing of the paper isn't doing it justice. In the Results section under "Identification of the GR gene PxylGr34 in P. xylostella" they would benefit from framing it in a question driven way, i.e. they were interested in searching for candidate Grs that may be involved in host selection within a model herbivore, and they looked for candidates among those that had been annotated from the P. xylostella genome. They then looked specifically for candidates that were highly expressed, comparing several chemosensory organs. Intriguingly, PxylGr34 was highly expressed in the antennae, as well as in other gustatory tissues, relative to other Grs. Because the expression data across Grs provides the biggest piece of evidence for focusing on this Gr, I would pull Figure 1—figure supplement 2 out from the supplement and stick it in Figure 1 as an additional panel. That being said, just because it is highly expressed, doesn't mean that it is the most important Gr for enabling feeding discrimination on their host plants – it would certainly be interesting to do the same experiments with all the other Grs but that's out of the scope of this paper for now.

Thank you so much for your suggestions. Accordingly, we reframe the Results section under “Identification of PxylGr34, a highly expressed GR gene in *P. xylostella*” in a question driven way. We also pull Figure 1—figure supplement 2 out from the supplement and make it as Figure 2A in the revised version.

2) "Brassinolide (BL) is a C28 brassinosteroid (BR)…" this whole paragraph needs to be restructured so that it's clearer why they focused on brassinolides, a plant hormone, rather than the typical anti-herbivore plant defenses, as mentioned in the previous sentence, or rather than other plant hormones. They outline some interesting reasons, i.e. their ubiquity in the plant kingdom, their structural similarity to molting hormones, but more explanation would benefit their study-there is still a way to frame this that makes the choice to focus on BLs clearer. After reading the rest of the paper, it's clear now that they tested 24 different phytochemicals of various groups, and brassinolide was the one that activated PxylGr34. But it makes sense for the Introduction to discuss more generally what their initial thoughts or hypotheses were about the kinds of compounds that should be important for herbivorous insects. If they had an a priori reason to be interested in brassinolides, then discuss that here; otherwise, leave it for the Discussion section.

We accept these good suggestions and add the related information in the revised Introduction. We move the paragraph about BL information to the Discussion.

3) Subsection “BL and its analog EBL induced a strong response in the oocytes expressing PxylGr34” – it would be great to mention earlier in this paper that 24 phytochemicals were tested, in the Abstract and Introduction. It would have answered my earlier questions about why there was a focus on brassinolides. Also, it might be worth mentioning that brassinolides might not be the only stimulant of Gr34; although 24 is a rather large number of compounds to test, there still may be others. Also, I only see the chemical listed in Figure 2. It would be great to list them at least in the Materials and methods, with source information.

We revise as “Functional analyses using the *Xenopus* oocyte expression system and 24 diverse phytochemicals showed that PxylGr34 is tuned to the canonical plant hormones brassinolide (BL) and 24-epibrassinolide (EBL).” in the Abstract. We also add several sentences in the Introduction as follows: “The chemical components in leaves of *Brassica* species, including sugars, sugar alcohols, amino acids, amines, glucosinolates and plant hormones, may be involved in such a process. Among these compounds, sinigrin and brassinolide (BL) have relatively higher concentrations in *Brassica* than in many other plant species (Fahey et al., 2001; Lv et al., 2014). Sinigrin has been proved as a feeding/ oviposition stimulant for *P. xylostella* (Gupta and Thorsteinson, 1960). The medial sensilla styloconica in the maxillary galea of *P. xylostella* larvae contain a GSN sensitive to sinigrin and other glucosinolates (van Loon et al., 2002). BL as a ubiquitous plant hormone has been widely studied in plant growth and development (Clouse and Sasse, 1998), but little is known about its behavioral effects on phytophagous insects.”

We add “It is worth pointing out that although the tested 24 compounds are a rather large compound profiles, the ligands of PxylGr34 could be more than BL and EBL. Given the high expression of *PxylGr34* in the taste organs of *P. xylostella*, we could not rule out the possibility that this gene also functions together with other *GRs*.” in the Discussion.

We add “Each of 24 compounds listed in Table 1 was diluted and the pH was adjusted to 7.5 in Ringer’s solution before being introduced to the oocyte recording chamber using a perfusion system.” in the Materials and methods, and list the tested 24 phytochemicals with source information in the revised Table 1.

4) Discussion paragraph six – it would make sense if this last paragraph or some part of the discussion was framed around why insects would want to detect brassinolides. That's the main question I've had throughout this paper. Is there evidence that insects routinely detect plant hormones to regulate their behavior (I know this to be true, but a paragraph on this topic csould be important)? Do these hormones correlate well with the nutritional content, growth, stress, defense concentrations, etc. within the plants to provide a reliable cue for where to feed? From there, you can then speculate that perhaps the BL may influence insect development because of the structural similarity with ecdysteroids, and then that discussion flows well from there.

We accept this suggestion. We add a new paragraph in Discussion as follows.

“Plant hormones, although generally found in small amounts and rarely toxic, play a key role in regulating plant growth, development, and resistance to biotic and abiotic stresses (Bari and Jones, 2009; Krouk et al., 2011; Wu and Baldwin, 2010). […] However, how the caterpillars eavesdrop the hormone signals remains a mystery. The present study provides the first evidence that *P. xylostella* could detect the plant hormone BL with a bitter gustatory receptor, which reflects a new adaptation of insects to plant defenses.”

5) Subsection “Phylogenetic analysis” – neighbor joining (NJ) method – NJ is not a true phylogenetic method per se (it ignores character information completely), and so I suggest you repeated this analysis by using maximum likelihood inference or Bayesian inference (BI). NJ is based on genetic distances, whereas ML or BI uses the character information and a model of sequence evolution to explore the tree space before selecting the most likely tree. The general consensus is that more robust phylogenetic methods should be used for datasets like these.

We accept your suggestion and repeat the phylogenetic analysis using the maximum likelihood inference. The original Figure 1—figure supplement 1 is updated as a new figure (Figure 1) in the revised version based on the new phylogenetic analysis. The related sentences about the phylogenetic method was also revised in Materials and methods.

6) Figure 1 and Figure 3: For Figure 1, I'd like to see all datapoints overlaid on each histogram. Same with Figure 3B and 3D and Figure 4, 5, 6 where practicable. All of the raw data should be deposited in an excel sheet that is easily accessible, so we can link each figure to a dataset readily (data for each figure on a different sheet in same file).

We accept the suggestion and show all datapoints overlaid on each histogram in the related Figures (Figure 2, Figure 4B, 4D, 4F, and Figure 5, Figure 6, Figure 6—figure supplement 1, Figure 7, 8, and 9 in the revised version). All of the raw data are provided as Source data files in the revised version.

Reviewer #2:The manuscript describes the expression pattern of gustatory receptors in taste organs of larvae and females of the cabbage moth. One of these receptors (GR34) belonging to the bitter receptors was found to have an especially high expression level. Heterologous expression of GR34 in *Xenopus* oocytes revealed a ubiquitous plant hormone, brassinolide, as the best ligand. Brassinolide turned out to be deterrent for foraging larvae and egg laying females. After silencing the expression of GR34 in larvae, the deterrent effect of the plant hormone was abolished. Although the manuscript adds valuable information about the function of gustatory receptors in insects, several points are not clearly enough explained to reach a broader audience.1) Novelty of the research: The authors stress in the Introduction that the molecular basis of deterrent gustatory neurons in phytophagous insects has not been investigated yet. However, one publication they cite at the end of their Discussion (Kasubuchi et al., 2018) could deorphanize several bitter GRs in the silk moth, and show the deterrent effect of a ligand of these GRs in larvae. It would be reasonable to mention this study already in the Introduction, and state in which way the present manuscript wants to add further insights, e.g. by silencing the receptor and testing oviposition behavior.

Thank you for your suggestion. We accept it and cite the mentioned study as “Most recently, BmorGr16, BmorGr18, and BmorGr53 showed response to coumarin and caffeine in vitro, and the coumarin had feeding deterrent effect on *B. mori* larvae (Kasubuchi et al., 2018); …” in Introduction.

We rephrase the related statement in the revised version as follows:

“In order to uncover the molecular basis of *P. xylostella* perceiving feeding/ oviposition stimulants and deterrents, we re-examined all the GRs reported from the previous studies of *P. xylostella*. Through transcriptome analysis and qPCR, we identified one bitter GR (PxylGr34) highly expressed in the larval head and the adult antennae.

Subsequently, we functionally analyzed this GR with *Xenopus* oocyte expression system and RNAi, and found that PxylGr34 is tuned to BL as a feeding and oviposition deterrent of *P. xylostella*.”

2) Role of the plant hormone brassinolide: It should be explained why it is interesting to test if insects can detect this ubiquitous plant hormone. Is brassinolide regularly tested in insect gustatory research? What kind of information might high doses of brassinolide that are necessary to elicit physiological and behavioral responses convey for the hungry larvae or egg-laying female? As the hormone is present across the plant kingdom and in almost every plant tissue it is difficult to imagine how it could be used to identify host plants.

Thank you for your suggestions. We add the information in the Introduction as follows:

“The chemical components in leaves of *Brassica* species, including sugars, sugar alcohols, amino acids, amines, glucosinolates and plant hormones, may be involved in such a process. Among these compounds, sinigrin and brassinolide (BL) have relatively higher concentrations in *Brassica* than in many other plant species (Fahey et al., 2001; Lv et al., 2014). Sinigrin has been proved as a feeding/ oviposition stimulant for *P.*

*xylostella* (Gupta and Thorsteinson, 1960). The medial sensilla styloconica in the maxillary galea of *P. xylostella* larvae contain a GSN sensitive to sinigrin and other glucosinolates (van Loon et al., 2002). BL as a ubiquitous plant hormone has been widely studied in plant growth and development (Clouse and Sasse, 1998), but little is known about its behavioral effects on phytophagous insects.”

As far as we known, brassinolide (BL) was not tested in previous insect taste studies although BL shows agonistic activity with 20hydroxyecdysone in many insect species (Zullo and Adam, 2002).

This study shows that the hungry larvae avoid feeding and the egg-laying females avoid ovipositing on the plant with the high concentration of BL. We suggest that this is an adaptation of this insect species to potential toxic substances in plants because BL has a similar structure with insect molting hormones such as 20-hydroecdysone and may have detrimental effect on survival and development of *P. xylostella*.

The plant hormone BRs are present across the plant kingdom and in almost every plant tissue, but their content differs widely among plant 4 species. For example, the concentration of BL is 1.37×10^-4^ g/kg in *Brassica campestris* L. leaves, while it is 1.25×10^-6^ g/kg in *Arabidopsis thaliana* leaves (Lv et al., 2014). Our results show that the threshold concentration of BL for behavioral inhibition of *P. xylostella* is in the range of 10^-4^ –10^-3^ g/kg, so we suggest that *P. xylostella* could detect the plants or plant tissues with higher concentration of BL with this receptor.

3) Molecular work: A) How were the 21 GRs analyzed in Figure 1—figure supplement 2 selected from the 42 bitter GRs identified the phylogenetic tree? B) Why is there a difference in the expression pattern of GR34 in female tissue in Figure 1-S3 (antenna, head and foreleg) and in Figure 1 (only antenna)?

We analyzed the expression of all the 67 validated GRs of *P. xylostella*, by calculating the transcripts per million (TPM) values based on our transcriptome data. However, 46 GRs showed very low expression and TPM was zero, so we only list the 21 GRs. We add “The GRs that undetectable in the TPM analysis were not listed.” in the figure legend.

Generally, most genes showed consistent results between RNAsequencing and qPCR data. However, the inconsistent expression results would be observed between them especially when the genes were lower expressed in some tissues (Everaert et al., 2017). This fits the case of Gr34 here. In this study, the expression of Gr34 in the head and foreleg are much lower than in the antennae although Gr34 also has some expression in female head and foreleg based on the qPCR analysis.

4) Effect of GR34 in egg laying behavior: The focus of the study was on larvae; however, it would be worth exploring the role of GR34 in female moths. Would it be possible to record from antennal gustatory sensilla of the female, and to silence GR34 also in females and investigate the effect on oviposition? It should be mentioned in the Introduction or Discussion that Plutella females touch the leaf surfaces with their antenna before egg laying to explain why gustatory receptors on the antenna might be useful for host plant choice.

Thank you for your suggestions. We agree that it would be worth exploring the role of Gr34 in female moths. However, the gustatory sensilla in the female moth antenna are very small and dense, we had tried but it is very hard to get recordings from the single gustatory sensillum from female antennae.

We totally agree that it would be very meaningful to silence Gr34 also in females and investigate the effect on oviposition. We had tried to inject the siRNA into the female pupae, but all failed because the pupae of *P. xylostella* is so small that the injection harmed the pupae severely. Therefore, we only successfully carried out the larval RNAi experiment and measured physiological and feeding effects in this work.

Sure, *P. xylostella* uses antennae to pat leaf surfaces before egg laying. We add “*P. xylostella* mainly selects *Brassica* species as its host plants, and its females pat the leaf surfaces with their antennae before egg laying (Qiu et al., 1998).” in Introduction.

Reviewer #3:In this manuscript, the authors characterized PxylGr34 of P. xylostella. They identified the full-length coding sequencing of PxylGr34 based on public data and their unpublished RNA-seq data. They found that PxylGr34 is highly expressed in larva head as well as female antennae in a real-time PCR. Through functional studies, they show that PxylGr34 is specifically tuned to BL and EBL in *Xenopus* oocyte in vitro system. They further show that BL evoked electrophysiological response in the medial sensilla styloconica on maxillary galea of larvae. Finally, they show BL inhibited larvae feeding and female oviposition of P. xylostella.The authors have done a good job of characterizing PxylGr34 in vitro analysis. However, this manuscript has to remedy several limitations to be published in eLife.1) It will be great if they can show the molecular properties of PxylGr34- for example, whether they are ligand-gated channels or G-protein coupled receptors.

Thank you for your suggestion. The molecular properties of GRs are a very important aspect in GR studies. Insect GRs are evolutionarily related to the insect olfactory receptors (ORs). The insect ORs have an inverted topology relative to GPCRs, and functions both in ligand-gated channels and G-protein coupled (Sato et al., 2008; Wicher et al., 2008). Similarly, insect GRs also have an inverted topology relative to GPCRs (Xu et al., 2012; Zhang et al., 2011), and may function in ligandgated ion channel (Sato et al., 2011). Given the close relationship between insect GRs and ORs, it could be that PxylGr34 functions both in ligandgated channel and G-protein coupled. We would further analyze the molecular properties of PxylGr34, and study its structure and function relationship in the next.

2) There are several issues with tip-recording.i) It is not clear whether PxylGr34 is required for BL-evoked in vivo electrophysiological responses. A knockdown experiment is recommended.

We accept your suggestion. We carried out a new knockdown experiment and tested the electrophysiological responses, which result in the new figures, Figure 8B and 8C. We found that the frequency of spikes to BL elicited in the medial sensilla styloconica of the PxylGr34 siRNA treated larvae was decreased. We add this result in revised version.

i) Representative traces are hard to read, making it difficult to determine whether these are actual spikes upon BL stimulation. Also, please add a y-axis scale bar.

Good suggestions and we accept both of them. We rearrange the representative traces of BL stimulation in the revised Figure 4A; add the y-axis scale bar in the revised Figure 4A, 4C, 4E, and Figure 5A, Figure 8B.

iii) Since it has been reported that Sinigrin and glucosinolates stimulate the same sensilla, it is beneficial to show the spikes evoked by these chemicals as a control.

It is a very good suggestion and we accept it. We carried out a new electrophysiological experiment using sinigrin as a positive control. New data are present in two new figures, Figure 4E and 4F. As previously reported, the medial sensilla styloconica also exhibited vigorous responses to sinigrin. However, the spike amplitudes induced by sinigrin were larger than those induced by BL and EBL (Figure 4E and 4F). These results suggest that BL and EBL activate the same neuron, while sinigrin activates a different neuron in the sensillum. We add this result in the revised version.

iv) Please show whether EBL can also stimulate the GSNs in the medial sensilla.

We accept your suggestion and run a new experiment to test the electrophysiological responses of the GSNs in the medial and lateral sensilla to EBL. The results show that of the two pairs of sensilla styloconica in the maxillary galea of 4^th^ instar larvae, the lateral sensilla styloconica had no response to EBL (Figure 4C and 4D); the medial sensilla styloconica exhibited vigorous responses to EBL at 3.3×10^-4^ M, and the spike amplitudes induced by BL and EBL were about the same (Figure 4A, B, C, and D). The spike amplitudes induced by sinigrin were larger than those induced by BL and EBL (Figure 4E and 4F). These results suggest that BL and EBL activate the same neuron, while sinigrin activates a different neuron in the sensillum. We add this result in the revised Figure 4C and 4D, and in the text.

v) It will be beneficial to show the schematic of larval taste organs.

It is a very nice suggestion, we accept it and add the schematic of larval taste organs in the revised Figure 4.

3) The effect of EBL is not convincing. It will be helpful to provide a dose-response profile with a wide range of EBL.

We accept your suggestion and run a new experiment. In the revised version, we provide a dose-response profile with a wide range of EBL in the Figure 6—figure supplement 1. We found that the feeding areas of larvae were significantly smaller on the leaf discs treated with EBL at concentrations of 10^-4^ M and above than on the control leaf discs, which is similar with BL. We add this result in the revised version.

4) It is hard to tell if the knockdown of PxylGr34 has an effect on feeding behavior. It seems that only control food feeding increased, in contrast to BL-containing food.

It is an important question, and then we repeated the treatment of the knockdown of PxylGr34 on the feeding behavior of 4^th^ instar larvae. The results show that the PxylGr34 siRNA-treated larvae had no significant preference for control leaf discs over those treated with BL, which is consistent with the result in the original manuscript. It proves that the knock-down of PxylGr34 by RNAi abolishes BL-induced feeding inhibition, which is not caused by control food feeding increased.

5) It is interesting whether PxylGr34 is only expressed in adult female antenna and involved in ovipositing. Is it expressed in male antenna? If it is, what is the role of PxylGr34 in male?

It is a nice question. We newly tested the expression of PxylGr34 in the male antennae, and found that PxylGr34 also has high expression in male antennae. For *P. xylostella* is a typical specialist and mate detection is closely related with the chemical cues of the host plants, we speculate that PxylGr34 may take part in the judgement of host plants in both male and females. We add this result in the revised Figure 2C, and in the text.

6) There is no description for Figure 2D in the main text.

Sorry. Now we add the description for it (now Figure 3D) in revised version.

7) The authors have to provide the experimental condition and data of the survival test with BL.

Thank you for your suggestion. The experimental condition is provided in Materials and methods as “The axisymmetric pinnate leaf was freshly picked from 4-week-old pea plants grown in a climate-controlled room. One leaf was folded in half, and two leaf discs (diameter, 7 mm) were punched from the two halves as the control (C) and treated (T) discs, respectively. For the treated discs, 5 µL (13 µL/cm^2^ ) of the test compound diluted in 50% ethanol was spread on the upper surface using a paint brush. For the control discs, 5 µL 50% ethanol was applied in the same way. Control (C) and treated (T) discs were placed in a C-T-C-T sequence around the circumference of the culture dish (60 mm diameter × 15 mm depth; Corning, NY, USA). After the ethanol had evaporated (15 min later), a single 4^th^ instar caterpillar (day 1), which had been starved for 6 h, was placed in each dish. The dishes were kept for 24 h at 23°–25°C in the dark, to avoid visual stimuli. Each dish was covered with a circular filter paper disc (diameter 7 cm) moistened with 200 µL ddH_2_O to maintain humidity.”

Considering BL did not affect larval survival but may reduce development time or have other detrmental effects because of its simularity with ecdysone, we delete the survival test with BL in the revised version, as suggested by Reviewer 1.

8) Typo Figure 1 legend “4th instar larvae” to “adult female.”

Sorry for this mistake. We correct it in the revised version.